# HubRouter: Learning Global Routing via Hub Generation and Pin-hub Connection

**Xingbo Du, Chonghua Wang, Ruizhe Zhong, Junchi Yan**[*]
Dept. of Computer Science and Engineering & MoE Key Lab of AI, Shanghai Jiao Tong University
`{duxingbo, philipwang, zerzerzerz271828, yanjunchi}@sjtu.edu.cn`

## Abstract

Global Routing (GR) is a core yet time-consuming task in VLSI systems. It recently attracted efforts from the machine learning community, especially generative models, but they suffer from the non-connectivity of generated routes. We argue that the inherent non-connectivity can harm the advantage of its one-shot generation and has to be post-processed by traditional approaches. Thus, we propose a novel definition, called **hub**, which represents the key point in the route. Equipped with hubs, global routing is transferred from a pin-pin connection problem to a hub-pin connection problem. Specifically, to generate definitely-connected routes, this paper proposes a two-phase learning scheme named HubRouter, which includes 1) **hub-generation phase**: A condition-guided hub generator using deep generative models; 2) **pin-hub-connection phase**: An RSMT construction module that connects the hubs and pins using an actor-critic model. In the first phase, we incorporate typical generative models into a multi-task learning framework to perform hub generation and address the impact of sensitive noise points with stripe mask learning. During the second phase, HubRouter employs an actor-critic model to finish the routing, which is efficient and has very slight errors. Experiments on simulated and real-world global routing benchmarks are performed to show our approach's efficiency, particularly HubRouter outperforms the state-of-the-art generative global routing methods in wirelength, overflow, and running time. Moreover, HubRouter also shows strength in other applications, such as RSMT construction and interactive path replanning.

## 1 Introduction

As the scale of integrated circuits (ICs) increases rapidly, the quality and efficiency of current Electronic Design Automation (EDA) technologies are being ceaselessly challenged. Among the various tasks from logic synthesis [50] to placement and routing [7], global routing (GR) [8, 24, 36, 43, 31, 35, 6, 48] is one of the complex and time-consuming combinatorial problems in modern Very Large Scale Integration (VLSI) design. As a posterior step of component placement, it generates routing paths to interconnect pins of IC components from a netlist, which are already placed on the physical layout [6]. The objective of global routing is basically to minimize total wirelength while avoiding congestion in the final layout[2]. However, even its simplified 'two-pin' case (see Fig. 8 in Appendix C for illustration) that routes every net with only two pins under design constraints turns out to be an NP-complete problem [27].

---

[*]Correspondence author. This work was partly supported by China Key Research and Development Program (2020AAA0107600), NSFC (62222607) and SJTU Scientific and Technological Innovation Funds.

[2]Another routing task called detailed routing [3] is orthogonal to this work. To our knowledge, there is so far no learning-related work on problems whose scale is even much larger than global routing (see example in [6]).

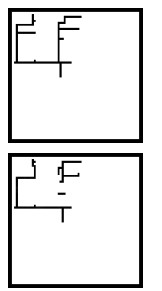

Figure 1: Example of an unconnected route. Up: Original route; Down: Generated route.

Table 1: Characteristics of global routing approaches. Classical and RL-based methods generally transfer routing into 2-pin problems. In particular, [31] is highly sensitive to chip scales. The generative model PRNet can generate each route in one-shot but cannot guarantee connectivity which requires considerable post-processing for failed routes. HubRouter in this paper has all the advantages of PRNet and in particular ensures connectivity.

| Model | Type | Multi-pin | Connectivity | Scalability |
|---|---|---|---|---|
| BoxRouter [8] | Classical | ✗ | ✓ | ✗ |
| DRL [31] | RL | ✗ | ✓ | ✗ |
| PRNet [6] | Generative | ✓ | ✗ | $\Delta^{*}$ |
| HubRouter (ours) | Generative | ✓ | ✓ | ✓ |

$^{*}$ Scalable for one-shot generation, but not scalable for post-processing.

Traditional works [8, 24, 36, 43] adopt (strong) heuristics to greedily solve global routing. However, the diversity and scale could pose new challenges to classical algorithms, which call for strategy updating and improvement by human experts on a continuous basis. To mitigate the reliance on manual efforts and facilitate the overall design automation and quality, machine learning has been adopted for global routing, as one of its diverse applications in chip design ranging from logic synthesis [40, 39] to placement [32, 28], etc. Specifically, deep reinforcement learning (DRL) [31, 35] and generative models [48] have been adopted to tackle global routing (sometimes also along with other tasks, e.g., placement [6] in the design pipeline). However, DRL methods suffer from large state space and often need to spend enormous time on generating routes as the scale of grids increases on the test instance, i.e., the netlist, which is practically intimidating for real-world global routing. The generative approaches can be more computationally tractable due to the potential one-shot generation capability and train/test the model on different instances. In fact, the generative models have recently been adopted in different design tasks [5, 56] beyond EDA (partly) for its higher efficiency compared with the iterative RL-based decision-making procedure, while a common challenge is how to effectively incorporate the rules and constraints in the generative models. However, though generative global routing approaches [6] inject connectivity constraints into the training objective, they often degenerate to an exhaustive search in post-processing when the generated initial routes fail to satisfy connectivity, as shown in Fig. 1. Our experimental results will show that the routes for difficult nets have a very low average connectivity rate of less than 20%, which means that over 80% generated routes for difficult nets require time-consuming post-processing. This greatly harms the inference time and indicates a serious challenge for the routing problem.

To address this intractable issue, our main idea is to propose a novel definition, '**hub**', which means the (virtual) key point in the route. By transferring the pin-pin connection problem to the hub-pin connection problem, situations where routes are unconnected in generative approaches can be avoided.

Specifically, we propose a novel two-phase learning scheme for global routing, named HubRouter, which includes 1) **hub-generation phase**: A condition-guided hub generator using deep generative models under a multi-task learning framework; 2) **pin-hub-connection phase**: A post-processing rectilinear Steiner minimum tree (RSMT) construction module that links the hubs using an actor-critic model. In the generation phase, hubs, routes, and stripe masks (a practical module as illustrated in Sec. 3.1) are together generated under a multi-task framework by generative models with optional choices, including Generative Adversarial Nets (GAN) [11], Variational Auto-Encoder (VAE) [26], and Diffusion Probabilistic Models (DPM) [16]. Though only hubs are required as outputs in this phase, we find it helpful to simultaneously generate routes and stripe masks together with hubs, where routes are used to obtain the local perception and stripe masks are capable of removing noise points. In the connection phase, we regard the connection of generated hubs as an RSMT construction problem, which is NP-complete [10]. Equipped with an actor-critic model [47], this phase can be conducted in a more scalable way [33] with very slight errors. We also introduce a special case that when hubs in the first phase are correctly generated for an RSMT route, its reconstruction time complexity can be reduced to $O(n \log n)$, which shows the scalability potential of HubRouter.

With this two-phase learning scheme, the proposed HubRouter enforces all generated routes to be connected and save the post-processing time in generative approaches [6]. Apart from the strength in global routing, HubRouter is of generality to deal with other applications. We evaluate HubRouter in

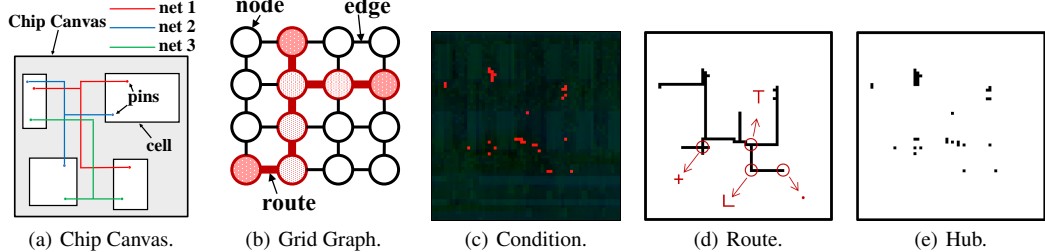

| (a) Chip Canvas. | (b) Grid Graph. | (c) Condition. | (d) Route. | (e) Hub. |

Figure 2: Diagrams of a) chip canvas; b) grid graph; c) condition image, with pin positions and capacity; d) route, and examples of 4 types of hubs; e) all hubs generated from d).

RSMT construction and interactive path replanning. Table 1 compares different routing methods and **the highlights of this paper are:**

1) We empirically show that the recent generative global routing approaches could suffer from the inherent non-connectivity of generated routes. Inspired by this observation, this paper proposes a new concept, namely **hub**, which means a key point in the route. With hubs generated for each net, pins can be connected efficiently under connectivity assurance.

2) Equipped with the concept of hubs, we devise HubRouter, which is a novel two-phase learning scheme for global routing, including (virtual) hub generation and pin-hub connection. Compared with generating routes directly, this scheme ensures the connectivity of all generated routes by generating intermediate hubs and then connecting them with pins. The connectivity assurance alleviates the time-consuming post-processing.

3) In the generation phase, we introduce a condition-guided generative framework under the perspective of multi-task learning, where routes and stripe masks are generated together with hubs to improve hubs' generation quality. In the connection phase, we adopt an actor-critic network to effectively simulate the RSMT construction. Moreover, we illustrate that when hubs are correctly generated for an RSMT route, the reconstruction can be optionally performed in $O(n \log n)$ time complexity.

4) Multiple experiments on simulated and real-world datasets are performed to illustrate the effectiveness of HubRouter. Especially, HubRouter ensures the connectivity of generated routes and outperforms the state-of-the-art generative global routing models in wirelength, overflow, and time. We also introduce two applications other than global routing to show the generality of HubRouter.

## 2 Background and Preliminaries

**Global Routing.** In VLSI design, global routing is a stage after placement, which determines the paths for nets and interconnects the pins on a chip layout. Typically, given a physical chip and a netlist, we have a chip canvas and several nets (see Fig. 2(a)), where each net includes some pins on the fixed positions decided by macro/standard cells in the previous placement process. In global routing, the chip canvas is further divided into rectangular tiles, where the tiles are formed into a grid graph $\mathcal{G}(\mathcal{V}, \mathcal{E})$ (see Fig. 2(b)). Nodes $\mathcal{V} = \{v_i\}_{i=1}^{|\mathcal{V}|}$, also named global routing cells (GCells), represent the tiles and we need to link all the tiles that contain pins. The nodes containing pins are dyed dark red in Fig. 2(b). Edges $\mathcal{E}$ represent the paths among adjacent nodes, where each edge $e_{ij} \in \mathcal{E}$ has its given capacity $c_{ij}$ and usage $u_{ij}$. The main objective of global routing is to connect all the required connections and on this basis, reduce the routing wirelength (WL) and overflow (OF). The overflow here refers to $o_{ij} = \max(0, u_{ij} - c_{ij})$, i.e., the exceeded number of routes compared to the given capacity on a tile.

**Generative Global Routing.** Actually, global routing is a combinatorial problem and can be formulated as a 0-1 integer linear programming (0-1 ILP) problem [4], but it is still NP-complete. Despite that various works [24, 8, 31] denote the layout of global routing as a routing graph, generative models [6] give a novel insight that regards the layout as an image and treats route generation as independent conditional image generation. Specifically, pixels are used to represent tiles in global routing, and a routing output image is generated from a given condition image. The condition image contains three channels, with the first channel being the locations of pins and the other two channels

Figure 3: Pipeline of the proposed two-phase scheme. 1) *hub-generation phase*: Route, hub, and mask are fed to three optional generative models, i.e., GAN, VAE, and DPM, with condition guided. The generated hub image is further masked by a stripe mask module; 2) *pin-hub-connection phase*: The hubs generated in the first phase are fed to an actor-critic network to obtain its RSMT construction by learning Rectilinear Edge Sequence (RES), and finally the connected route is obtained.

being the capacity of horizontal/vertical grid edges. The routing image is a grayscale image with a value of 255 representing routed and 0 representing unrouted. Examples of condition and routing images are respectively shown in Fig. 2(c) and 2(d).

**Hub.** Fig. 2(d) also introduces *hubs*, the main concept proposed in this paper, with 4 different types, which can be vividly represented as '$+, \top, \llcorner, \cdot$'. Formally, we have

**Definition 1** (**Hub**). *Given a one-channel image with $m \times n$ pixels, let $p_{ij}$ $(1 \leq i \leq m, 1 \leq j \leq n)$ denote the pixel in the $i$-th row and $j$-th column, whose value $r_{ij} = 1/0$ respectively represent routed/unrouted. Auxiliary denote $r_{0j} = r_{(m+1)j} = r_{i0} = r_{i(n+1)} = 0$, then the pixel $p_{ij}$ is a hub if and only if $r_{ij} = 1$ and it satisfies any of the following condition:*

(1) $+ : r_{(i-1)j} = r_{(i+1)j} = r_{i(j-1)} = r_{i(j+1)} = 1$;     (2) $\top : r_{(i-1)j} + r_{(i+1)j} + r_{i(j-1)} + r_{i(j+1)} = 3$;

(3) $\llcorner : r_{(i-1)j} + r_{(i+1)j} = 1$ and $r_{i(j-1)} + r_{i(j+1)} = 1$;   (4) $\cdot : r_{(i-1)j} + r_{(i+1)j} + r_{i(j-1)} + r_{i(j+1)} = 1$.

This means that any routed pixel is a hub unless it has exactly two opposite routed neighbor pixels. In Fig. 2(e), we give an example of all the hubs generated from Fig. 2(d).

**Difference between Hubs and Steiner Points.** The intuition of hubs is similar to the concept of Rectilinear Steiner Point (RSP) [21], but we claim that there are essential differences between them. RSPs are searched for a global minimum total distance, while hubs are used to determine a path. Obviously, RSPs are special cases of hubs, and hubs can generate paths of different shapes at will (not only the shortest). Furthermore, traditional approaches can obtain hubs only after accomplishing the routing, while the characteristics of machine-learning approaches naturally give them the capability of learning such hubs. We envision that HubRouter has more applications compared with RSPs. To show this, we also conduct some applications, including RSMT construction and interactive path replanning in Sec. 4.4. Related works in this work are presented in Appendix A.

## 3 From Pin-Pin to Hub-Pin Connection: A Two-phase Routing Scheme

**Approach Overview.** We formulate global routing as a two-phase learning model. The first phase is called the *hub-generation phase*, where we approximate the conditional distribution $p_\theta(\mathbf{x}|\mathbf{z}, \mathbf{c})$ to the prior distribution $p(\mathbf{x}|\mathbf{c})$ when given $\mathbf{z} \sim p_\mathbf{z}(\mathbf{z})$ and condition $\mathbf{c}$. Here $\mathbf{z}$ is a latent variable from a prior distribution (usually assumed as Gaussian distribution) while $\mathbf{c}$ and $\mathbf{x}$ are respectively condition and input images (route, extracted hubs, and stripe mask). The second phase is called the *pin-hub-connection phase*, where we link the hubs generated in the first phase to obtain the final route.

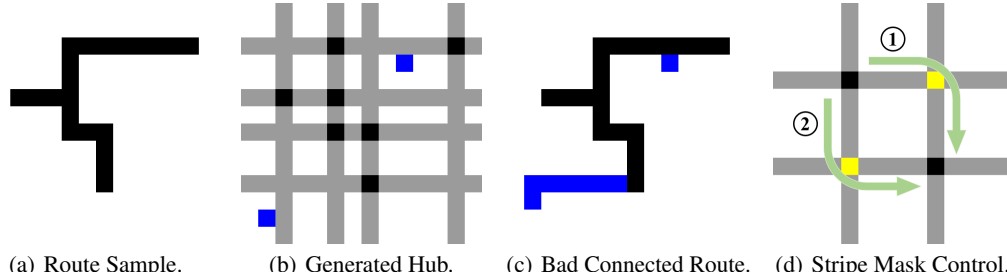

| (a) Route Sample. | (b) Generated Hub. | (c) Bad Connected Route. | (d) Stripe Mask Control. |

Figure 4: Illustration of how the stripe mask alleviates the noise impact. (a) A route sample. (b) The generated hubs, where the black/blue points are respectively hubs/noises. The gray stripes are generated masks. (c) Without the stripe mask, the route will be lengthened by some wrong and redundant paths (blue). (d) Two paths with the same length scheduled by the stripe mask.

This process can be treated as RSMT construction, and we follow REST [33] to use an RL-based algorithm to finish the routing. We respectively introduce these two phases in Sec. 3.1 and Sec. 3.2. In the generation phase, we also propose a multi-task learning framework to improve the quality of generated hubs, especially a novel stripe mask learning method is proposed to alleviate the negative effects caused by noise points cases. The overall algorithm is in Appendix B.

## 3.1 Hub-generation Phase: Multi-task Learning of Hub, Mask, and Unused Pre-Routing

Hub generation can be formulated as an image-to-image task because the prior distribution $p(\mathbf{x}|\mathbf{c})$ is given from training dataset $\{\mathbf{x}_i, \mathbf{c}_i\}_{i=1}^N$ with $N$ samples, where $N$ denotes the total number of nets for training. So, it seems that we can learn a distribution $p_\psi(\mathbf{x}|\mathbf{c})$ and can generate $\mathbf{x}' \sim p_\psi(\mathbf{x}|\mathbf{c} = \mathbf{c}')$ given a new condition $\mathbf{c}'$ out of the training dataset. However, since the results of hub generation are not unique, a generative probabilistic distribution $p_\theta(\mathbf{x}|\mathbf{z}, \mathbf{c})$ conditioned on $\mathbf{c}$ can better describe the routing. The main objective of hub generation is to minimize the difference between probability distributions $p(\mathbf{x}|\mathbf{c})$ and $p_\theta(\mathbf{x}|\mathbf{z}, \mathbf{c})$ while it often differs in expression under different perspectives. With the help of the rapid development of deep generative models, various works give us optional choices to address hub generation. In Appendix B, we respectively introduce how to incorporate GAN [11, 37], VAE [26, 44], and DPM [16, 17] into our generative framework.

Note that PRNet [6] has a similar generation model structure to ours, yet we fall into different plights. PRNet employs a bi-discriminator [54] in CGAN [37] to inject the connectivity constraints into the training objective. This constraint indeed works and increases the correctness (all pins are connected within one route) rate by over 10%, but it is almost useless for complicated cases, as will further be discussed in Sec. 4.2. Under our two-phase learning scheme, the connectivity can be guaranteed, but the generated noise points have a more negative impact on the final results. Thus, we propose a multi-task learning framework in the following to handle this potential weakness.

We claim that hub generation is highly different from other image generation missions because subtle noises can hardly affect their generation quality. When generating hubs, a noise point, especially the outermost one, can largely harm the wirelength of routing. We show this challenging phenomenon in Fig. 4(b) and 4(c), where two noise points (blue) in Fig. 4(b) lead to a much longer route in Fig. 4(c) since all the generated hubs should be connected within one route in our scheme. As discussed in some other special generation tasks like layout generation [5], high reconstruction accuracy is typically maintained during training to ensure the generation is suitable for its requirement. Following this setting, we further devise a multi-task learning framework, whereby the input in Sec. 3.1 is defined as $\mathbf{x} = \{\mathbf{x}^{(hub)}, \mathbf{x}^{(rt)}, \mathbf{x}^{(msk)}\}$ with three components, respectively hub, route, and mask. Note that although the hub generation is the main purpose in the first phase, we still generate routes and masks for auxiliary usages. In particular, the routes $\mathbf{x}^{(rt)}$ are pre-routing results and are unused in the second phase, which is employed to better obtain the continuous local perception in CNN-based networks. We further propose a novel mask learning module named *stripe mask* to focus on bad cases for hub generation. Specifically, the stripe mask is defined as a matrix $\mathbf{x}^{(msk)} \in \{0, 1\}^{m \times n}$, where $\mathbf{x}_{ij}^{(msk)} = 1$ means there is a hub in the $i$-th row or $j$-th column; otherwise $\mathbf{x}_{ij}^{(msk)} = 0$. At inference

time, the generated hubs are masked by a binarized stripe mask $\hat{\mathbf{x}}_{ij}^{(hub)} = \mathbf{x}_{ij}^{(hub)} \times \tilde{\mathbf{x}}_{ij}^{(msk)}$ with

$$\tilde{\mathbf{x}}_{ij}^{(msk)} = \mathbb{I}\left[\max\left(\frac{1}{m}\sum_{i=1}^{m}\mathbf{x}_{ij}^{(msk)}, \frac{1}{n}\sum_{j=1}^{n}\mathbf{x}_{ij}^{(msk)}\right) > \frac{1}{2}\right], \tag{1}$$

where $\mathbb{I}[\cdot]$ is an indicator function yielding $1/0$ if the input is correct/wrong. Eq. 1 shows that unless the noises in vertical/horizontal lines are dense, the stripe mask will not be formed. Fig. 4(c) shows that, with the help of the stripe mask, the blue noise points can be masked, and the generated route can be shortened by removing the redundant (blue) paths. Empirically, an appropriate stripe mask can eliminate most of these noise points. The reasons why we use the stripe mask rather than other masks [46] are due to the characteristics of global routing: Compared with other generation tasks, 1) Noise points will greatly impact the routing quality, and the stripe mask learning can eliminate most of them; 2) the routing result is not required to be unique while the stripe mask module is capable of providing different choices of routing. As shown in Fig. 4(d), two black hubs can choose an arbitrary yellow point as a turning point to connect each other while the length of paths is the same.

### 3.2   Pin-hub-connection Phase: RSMT Construction

After the hub-generation phase, the hub points will be sent to the pin-hub-connection phase to link all the generated hubs and obtain the final route. In this phase, the connection process is regarded as a rectilinear Steiner minimum tree (RSMT) construction problem, which is proved to be NP-complete [10]. Note that it seems to be possible to directly use RSMT construction to perform global routing; however, we claim that the hub-generation phase is essential because 1) Without the generated hubs, RSMT construction can only find the shortest route and fail to generate an appropriate route under restriction; 2) A good hub-generation phase for RSMT route reconstruction can be reduced to $O(n \log n)$, as stated in the following theorem:

**Remark 1.** *[Reconstruction Bonus] Suppose a set of hubs in an RSMT route is correctly generated, then its RSMT reconstruction can be performed in $O(n \log n)$ time complexity.*

We further illustrate it in Appendix C. Remark 1 introduces the optimal time complexity of the pin-hub-connection phase for the ideal cases. While without this strong assumption, we follow REST [33] to learn the Rectilinear edge sequence (RES) under an actor-critic neural network framework. Given a point set $\mathcal{P} = \{(x_i, y_i)\}_{i=1}^{|\mathcal{P}|}$, we want to get the RES, which is a sequence of $|\mathcal{P}| - 1$ index pairs $[(v_1, h_1), \ldots, (v_{|\mathcal{P}|-1}, h_{|\mathcal{P}|-1})]$ and pair $(v_i, h_i)$ means connecting point $v_i$ and $h_i$ with rectilinear edges. RES can be modeled as a sequential decision-making problem, so it is advantageous in RL. Note that the validity and existence of RES for the RSMT construction are already proved in REST.

As shown in Fig. 3, the actor network is used to create RES results with given coordinate points, while the critic network sets a baseline to predict the expected length of RSMT constructed by the actor network and guide the actor network to obtain better performance. According to the evaluation of the critic network, the actor network will update the policy to minimize the RSMT length. Specifically, the actor network is devised via an auto-encoder to learn better representations. We draw $B$ point sets $\bar{V} = \{V_1, V_2, \ldots, V_B\}$, and construct valid RES sets $R^{(V_i)}$ for each point set $V_i$. The objective is to minimize the expected advantage of the RSMT as follows:

$$\min_{\xi}\ \mathbb{E}_{V_i \sim \bar{V}, r_i \sim R^{(V_i)}}\left[b_{\zeta}(V_i) - b(V_i, r_i)\right] p_{\xi}(r_i), \tag{2}$$

where $b_{\zeta}(V_i)$ is a predicted length of RSMT by the critic network and $b(V_i, r_i)$ is the actual length evaluated in linear time. $p_{\xi}(r_i)$ is the learnable probability of generating a specific RES with regard to $\xi$. This objective is learned with an RL-algorithm [52]. Apart from the actor network, the critic one is learned simply by mean square error (MSE) with gradient descent training:

$$\min_{\zeta}\ \mathbb{E}_{V_i \sim \bar{V}, r_i \sim R^{(V_i)}}\|b_{\zeta}(V_i) - b(V_i, r_i)\|^2. \tag{3}$$

At test time, in line with [33], we apply 8 transformations that rotate the point set by [0, 90, 180, 270] degrees with/without swapping $x$ and $y$ coordinates and then choose the best solution. This trick is relatively more effective when hubs are not well generated.

Table 2: Experiments on ISPD-07 with 4 kinds of cases (detailedly introduced in Appendix D ). We compare HubRouter with PRNet [6] on three metrics. Optimal results are in **bold**.

| Metric | Case | PRNet (GAN) | HubRouter (VAE) | HubRouter (DPM) | HubRouter (GAN) |
|---|---|---|---|---|---|
| **Correctness Rate (%)** | Route-small-4 | 0.806 | $\mathbf{1.000}_{\pm\mathbf{0.000}}$ | $\mathbf{1.000}_{\pm\mathbf{0.000}}$ | $\mathbf{1.000}_{\pm\mathbf{0.000}}$ |
| | Route-small | 0.334 | $\mathbf{1.000}_{\pm\mathbf{0.000}}$ | $\mathbf{1.000}_{\pm\mathbf{0.000}}$ | $\mathbf{1.000}_{\pm\mathbf{0.000}}$ |
| | Route-large-4 | 0.196 | $\mathbf{1.000}_{\pm\mathbf{0.000}}$ | $\mathbf{1.000}_{\pm\mathbf{0.000}}$ | $\mathbf{1.000}_{\pm\mathbf{0.000}}$ |
| | Route-large | 0.040 | $\mathbf{1.000}_{\pm\mathbf{0.000}}$ | $\mathbf{1.000}_{\pm\mathbf{0.000}}$ | $\mathbf{1.000}_{\pm\mathbf{0.000}}$ |
| **Wirelength Rate (%)** | Route-small-4 | **1.001** | $1.099_{\pm0.020}$ | $1.060_{\pm0.011}$ | $1.011_{\pm0.003}$ |
| | Route-small | 1.009 | $1.042_{\pm0.006}$ | $1.174_{\pm0.009}$ | $\mathbf{1.002}_{\pm\mathbf{0.001}}$ |
| | Route-large-4 [*] | - | $1.122_{\pm0.039}$ | $1.100_{\pm0.021}$ | $\mathbf{1.005}_{\pm\mathbf{0.002}}$ |
| | Route-large [*] | - | $1.041_{\pm0.014}$ | $1.242_{\pm0.021}$ | $\mathbf{1.001}_{\pm\mathbf{0.000}}$ |
| **Generation Time (GPU Sec)** | Route-small-4 | 14.99 | $\mathbf{7.14}_{\pm\mathbf{0.19}}$ | $673.21_{\pm5.08}$ | $7.16_{\pm0.05}$ |
| | Route-small | 18.51 | $\mathbf{7.70}_{\pm\mathbf{0.20}}$ | $670.23_{\pm2.45}$ | $8.36_{\pm0.35}$ |
| | Route-large-4 | 19.47 | $\mathbf{7.24}_{\pm\mathbf{0.30}}$ | $673.01_{\pm5.18}$ | $7.53_{\pm0.16}$ |
| | Route-large | 19.22 | $10.65_{\pm0.09}$ | $672.86_{\pm4.44}$ | $\mathbf{9.75}_{\pm\mathbf{0.35}}$ |

[*] The correctness rate is too low for PRNet to reasonably evaluate the wirelength rate.

## 4 Experiment and Analysis

We introduce datasets and setups in Sec. 4.1. To show the performance of HubRouter, in Sec. 4.3, we respectively perform global routing on three kinds of datasets with both classical and learning baselines. Ablation study is performed to evaluate key modules in HubRouter. In Sec. 4.4, we introduce two applications, including RSMT construction and interactive path replanning. Finally, we analyze the time overhead and show the scalability of HubRouter. Each experiment in this section is run on a machine with i9-10920X CPU, NVIDIA RTX 3090 GPU and 128 GB RAM, and is repeated 3 times under different seeds with mean and standard deviation values in line with [6].

### 4.1 Datasets and Setups

**Datasets.** For training, we construct global routing instances by adopting NCTU-GR [34] to route on ISPD-07 routing benchmarks [38], which is in line with [6]. For test, we split the samples outside the training set into 4 types of routes according to the number of pins and the distance of pins. In detail, 'Route-small-4' and 'Route-small' respectively represent cases with no more than and more than 4 pins and the Half-perimeter wirelength (HPWL) of pins is less than 16. Here, HPWL = $[\max(x_i) - \min(x_i)] + [\max(y_i) - \min(y_i)]$ for a group of pin positions $\{(x_i, y_i)\}_{i=1}^{n_p}$ in a net with $n_p$ pins. 'Route-large-4' and 'Route-large' are similar, whose HPWL is more than 16. Moreover, we introduce ISPD-98 [1] routing benchmarks and some simulated small-scale cases used by a deep reinforcement learning (DRL) method [31], named DRL-8 and DRL-16.

**Metrics.** On ISPD-07 routing benchmarks, we choose correctness (all pins are connected within one route) rate and wirelength ratio (WLR, the ratio of the generated route length to the ground truth route length) introduced in [6] as our metrics. In the experiments of ISPD-98 and DRL cases, wirelength (WL), overflow (OF), and routing runtime are adopted. We also investigate some common metrics for generative models like Fréchet Inception Distance (FID) [15] and Inception Score (IS) [2, 42], but they are not suitable for taking on the real performance on global routing.

**Other Implementation Details.** Details of training/test datasets and other protocols, including introduction of baselines and model structures, are presented in Appendix D.

### 4.2 Unconnected Cases in Previous Generative Global Routing Approaches

We investigate the state-of-the-art generative global routing approach [6] and show that its correctness rate (CrrtR) is extremely low for cases with complex connection of pins. As declared in Table 2, the CrrtR is only 0.04 for the Route-large case. This is reasonable because connectivity is assured only when almost each pixel of routed path is correctly generated in the routing image, which is a very strict condition for complex cases. However, under our proposed two-phase learning scheme, HubRouter guarantees the CrrtR on all nets with the intermediately generated hubs. Moreover, despite

Table 3: Experiments on ISPD-98 (IBM01-06) and GRL cases. Wirelength (WL) and running time are compared among 4 baselines and HubRouter (brown) with generative structures (HR-VAE, HR-DPM, HR-GAN). Optimal results of WL and time are in **bold**. The results of DQN [31] on ISPD-98 are out of time (OOT) within 2 weeks, so only GRL cases are displayed.

| Metric | Model | IBM01 | IBM02 | IBM03 | IBM04 | IBM05 | IBM06 | GRL-8 | GRL-16 |
|---|---|---|---|---|---|---|---|---|---|
| **WL** | Labyrinth [24] | 75909 | 201286 | 187345 | 195856 | 420581 | 341618 | 2376 | 8204 |
| | Boxrouter [8] | 63687 | 172304 | 147463 | 169033 | **410614** | 280477 | 2328 | 7991 |
| | DQN [31] | OOT | OOT | OOT | OOT | OOT | OOT | 2434 | 8356 |
| | PRNet [6] | 61950 | 172802 | 152037 | 170493 | 420274 | 287777 | 2497 | 8172 |
| | HR-VAE | $64703_{\pm1498}$ | $176492_{\pm6830}$ | $159968_{\pm3281}$ | $179895_{\pm5274}$ | $434942_{\pm2916}$ | $300448_{\pm5560}$ | $2415_{\pm33}$ | $8584_{\pm244}$ |
| | HR-DPM | $66446_{\pm1586}$ | $190588_{\pm2337}$ | $168454_{\pm2486}$ | $183696_{\pm1736}$ | $475820_{\pm5516}$ | $316700_{\pm2843}$ | $\mathbf{2285_{\pm7}}$ | $\mathbf{7746_{\pm29}}$ |
| | HR-GAN | $\mathbf{61056_{\pm151}}$ | $\mathbf{167545_{\pm236}}$ | $\mathbf{147050_{\pm208}}$ | $\mathbf{164298_{\pm326}}$ | $411857_{\pm472}$ | $278198_{\pm423}$ | $2306_{\pm8}$ | $7768_{\pm30}$ |
| **Time (Sec)** | Labyrinth [24] | **6.47** | 10.14 | 12.07 | 36.81 | **7.54** | 18.43 | **< 1 second** | **< 1 second** |
| | Boxrouter [8] | 7.01 | 12.91 | 11.74 | 41.76 | 13.45 | 29.17 | **< 1 second** | **< 1 second** |
| | DQN [31] | OOT | OOT | OOT | OOT | OOT | OOT | > 1 day | > 1 day |
| | PRNet [6] | 254.31 | 585.23 | 523.34 | 573.19 | 1606.00 | 1227.31 | 11.54 | 28.25 |
| | HR-VAE | $9.66_{\pm0.08}$ | $\mathbf{9.69_{\pm0.04}}$ | $\mathbf{10.19_{\pm0.06}}$ | $\mathbf{12.93_{\pm0.07}}$ | $14.58_{\pm0.00}$ | $\mathbf{17.28_{\pm0.16}}$ | $5.83_{\pm0.01}$ | $5.88_{\pm0.02}$ |
| | HR-DPM | $1796.09_{\pm38.68}$ | $2772.29_{\pm16.83}$ | $2936.52_{\pm21.23}$ | $3865.21_{\pm25.07}$ | $4369.47_{\pm22.56}$ | $4965.08_{\pm121.46}$ | $37.00_{\pm2.58}$ | $53.90_{\pm2.85}$ |
| | HR-GAN | $41.02_{\pm0.51}$ | $46.58_{\pm0.56}$ | $52.04_{\pm2.35}$ | $67.31_{\pm3.51}$ | $72.28_{\pm3.72}$ | $88.02_{\pm4.45}$ | $6.31_{\pm0.20}$ | $6.38_{\pm0.04}$ |

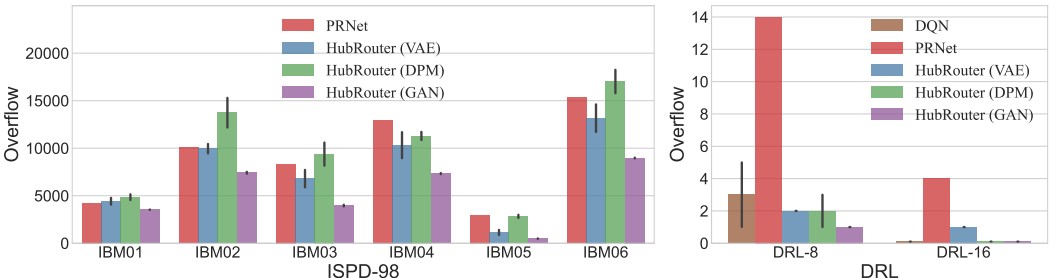

Figure 5: Initialized overflow on ISPD-98 (IBM01-06) and the DRL cases.

that PRNet only counts the wirelength of the connected routes, HubRouter (GAN) is competitive with it in WLR. Note that we only evaluate the inference time of generation in Table 2 (total time is analyzed in the next part), where PRNet is efficient with its one-shot superiority, though HubRouter (VAE/GAN) are still slightly better than it. An exception comes that HubRouter (DPM) suffers from time-consuming sampling for its multiple sampling steps.

### 4.3 Benchmarking Global Routing and Ablation Study

We test the WL, OF, and inference time on IBM01-06 from ISPD-98 as well as DRL cases. The detailed inference time that is divided into generation time and connection time is shown in Appendix E.4. As shown in Table 3, the WL of HubRouter (GAN) outperforms others on ISPD-98 cases while HubRouter (DPM) hits the optimum on DRL cases. This implies that HubRouter (DPM) is more capable on small-scale cases as the model complexity is not very high that easily incurs overfitting; however, due to its large inference time, we do not decide to increase its model complexity, but we believe it has high potential with more support of computing power. In addition, though HubRouter (VAE) fails to reach the top in WL, it is the fastest among all ML-based algorithms and is even competitive with classical algorithms in some cases. Since there exists the theoretical lower bound of this task [8], we also compare the relative error on ISPD-98 cases in Appendix E.3 to better demonstrate the performance of HubRouter. As these generative models empirically show different behaviors, we argue that when given a new dataset, the choice of generative algorithms is decided by the industrial needs. Specifically, if one needs high quality, GAN is the best. If speed is highly required, VAE is the best. If one has abundant resources, DPM can be adopted. Note that PRNet is significantly more time-consuming in this experiment compared with the results in Table 2 due to its post-processing. In line with PRNet [6], as the overflow is decreased by rerouting, we compare the initialized overflow among ML-based algorithms. As shown in Fig. 5, HubRouter-GAN has a minimal overflow in all test cases.

Ablation study is conducted in both generation and connection phases to show the influence of the introduced modules. As depicted in Table 4, in the hub-generation phase, we respectively denote the HubRouter with inputs of hubs, hubs + routers, and hubs + routers + stripe masks as HubRouter-h,

Table 4: Ablation Study on ISPD-07 with HubRouter (using GAN as generative embodiment).

| Generation-phase | Model | HubRouter-h | HubRouter-hr | HubRouter-hrm |
|---|---|---|---|---|
| | WLR | 1.031 | 1.022 | 1.005 |
| Connection-phase | Model | HubRouter-RMST | HubRouter (T=1) | HubRouter (T=8) |
| | WLR | 1.099 | 1.087 | 1.005 |

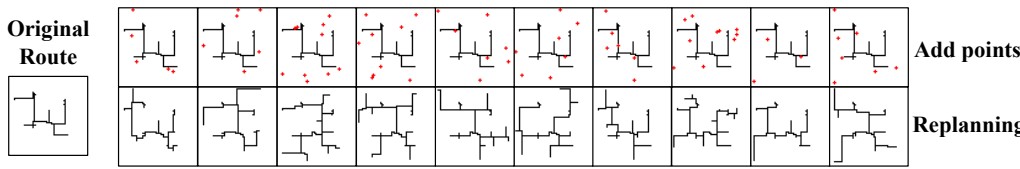

Figure 6: Samples of interactive path replanning. The first line is routes with some added points while the second line is the replanned paths.

HubRouter-hr, and HubRouter-hrm. Among all, HubRouter-hrm reaches the optimal wirelength rate and HubRouter-hr also outperforms HubRouter-h, which indicates that the multi-task learning with route and stripe mask is effective. In the pin-hub-connection phase, HubRouter-RMST is denoted as the one using R-MST as the connection model. In addition, HubRouter (T=1)/(T=8) respectively represent the HubRouter with/without 8 transformations introduced in Sec. 3.2. As expected, HubRouter (T=8) performs the best. Details on hyperparameter are in Appendix E.

### 4.4 Further Studies and Discussion

We envision that HubRouter can do more than just global routing. Here, we respectively perform RSMT construction and interactive path replanning to reflect the generality of HubRouter. The HubRouter in this section uses GAN as the generative embodiment due to its better performance.

**RSMT Construction.** It is intuitive that global routing for each net without overflow constraint is to minimize the wirelength, and thus is close to the problem of RSMT construction. A key question is whether the generated hubs can enhance the construction of the pure RSMT during the connection phase. To evaluate it, we test the average percent errors compared with GeoSteiner [51] on the random point data given by REST [33], together with running time, in Table 5. We compare baselines including R-MST [1], BGA [23], FLUTE [53] and REST [33]. When degree is smaller than 25 REST performs the best, but HubRouter outperforms others when degree is larger than 30. In particular, an unintuitive phenomenon shows that HubRouter performs even better with a higher degree. This is due to the characteristic of hub generation since a noise point in a dense case has much less impact on the error of wirelength than in a sparse case. Apart from the error, REST has the shortest running time depending on large batch size. However, HubRouter has to run sequentially in this task because degrees are different after generating some new hubs.

**Interactive Path Replanning.** Path replanning [19] refers to the process of revising a given path when the environment changes. It is a technique used in human-machine interaction. HubRouter is capable of deciding a new path with user-defined hubs. Fig. 6 shows samples of interactive path replanning results, where HubRouter connects new points and maintains most of the previous paths.

**Analysis of Scalability and Time Overhead.** We claim that HubRouter is scalable in both phases. In the generation phase, for a fixed scale of routing graph, similar to PRNet [6], the hub generation is one-shot. This implies that its generation speed is unrelated to the number of pins and is linear to the number of nets, as respectively shown in Fig. 7 (left/middle). For connection, we show in Sec. 3.2 that the RSMT construction can degrade to a $O(n \log n)$ R-MST problem under some ideal assumptions. In addition, as depicted in Fig. 7 (right), equipped with an actor-critic network, the time overhead of HubRouter increases more slowly than pure RSMT construction.

**Non End-to-End framework.** HubRouter is a two-phase framework that is designed with careful analysis. It differs from PRNet [6], which is an end-to-end attempt to produce the routes directly, but

---

[1] https://github.com/shininglion/rectilinear_spanning_graph

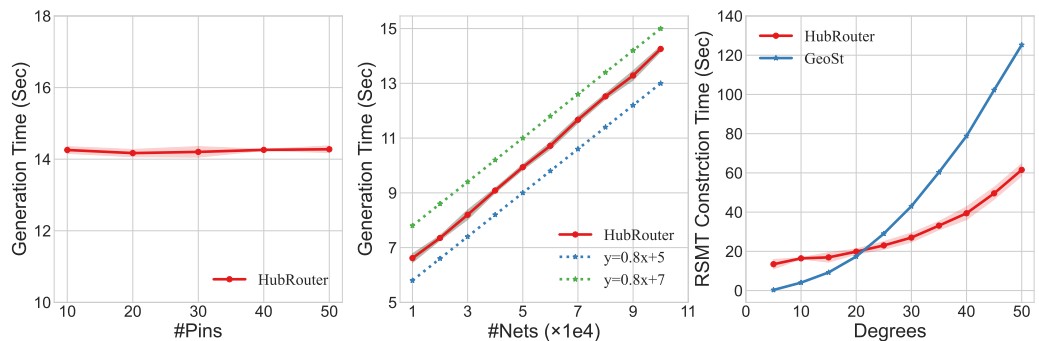

Figure 7: Generation time of HubRouter is constant to number of pins (left) and is linear to number of nets (middle). The RSMT construction time of HubRouter rises slower than GeoSteiner [51] (right).

Table 5: Experiments on RSMT construction. HubRouter and four baselines are compared to the RSMT solver GeoSteiner (GeoSt) and the best results on errors are in **bold**. Time overhead is also evaluated on GeoSt, REST, and HubRouter. Mean and variance are given with three trials.

| Degree | Average Percent Errors (%) | | | | | | Time (Sec) | | |
|---|---|---|---|---|---|---|---|---|---|
| | **GeoSt** | **R-MST** | **BGA** | **FLUTE** | **REST** | **HubRouter** | **GeoSt** | **REST** | **HubRouter** |
| **5** | 0.00 | 10.91 | 0.23 | **0.00** | **0.00** | $0.24_{\pm0.07}$ | $0.31_{\pm0.02}$ | $11.89_{\pm0.24}$ | $13.44_{\pm2.41}$ |
| **10** | 0.00 | 11.96 | 0.48 | 0.04 | **0.01** | $0.18_{\pm0.02}$ | $4.05_{\pm0.03}$ | $12.16_{\pm0.10}$ | $16.43_{\pm0.30}$ |
| **15** | 0.00 | 12.19 | 0.53 | 0.06 | **0.03** | $0.15_{\pm0.01}$ | $9.26_{\pm0.08}$ | $13.80_{\pm0.43}$ | $16.92_{\pm2.43}$ |
| **20** | 0.00 | 12.41 | 0.57 | 0.11 | **0.07** | $0.13_{\pm0.01}$ | $17.35_{\pm0.19}$ | $14.79_{\pm0.17}$ | $19.85_{\pm1.32}$ |
| **25** | 0.00 | 12.47 | 0.58 | 0.18 | **0.12** | $0.12_{\pm0.01}$ | $29.01_{\pm0.07}$ | $16.68_{\pm0.54}$ | $23.01_{\pm1.99}$ |
| **30** | 0.00 | 12.56 | 0.60 | 0.23 | 0.16 | $\mathbf{0.10}_{\pm0.01}$ | $43.00_{\pm0.17}$ | $18.53_{\pm0.29}$ | $27.02_{\pm2.30}$ |
| **35** | 0.00 | 12.63 | 0.62 | 0.26 | 0.21 | $\mathbf{0.09}_{\pm0.01}$ | $60.32_{\pm0.20}$ | $19.67_{\pm0.85}$ | $33.11_{\pm2.12}$ |
| **40** | 0.00 | 12.65 | 0.63 | 0.29 | 0.25 | $\mathbf{0.08}_{\pm0.01}$ | $78.80_{\pm0.06}$ | $20.91_{\pm0.51}$ | $39.42_{\pm3.26}$ |
| **45** | 0.00 | 12.67 | 0.63 | 0.30 | 0.32 | $\mathbf{0.07}_{\pm0.01}$ | $102.26_{\pm0.21}$ | $22.36_{\pm0.42}$ | $49.57_{\pm3.05}$ |
| **50** | 0.00 | 12.72 | 0.64 | 0.29 | 0.36 | $\mathbf{0.06}_{\pm0.01}$ | $125.26_{\pm0.41}$ | $23.79_{\pm0.20}$ | $61.55_{\pm3.07}$ |

faces the challenge of non-connectivity and inefficient post-processing. HubRouter addresses this by generating hubs in the middle stage and connecting them with pins to ensure connectivity. However, this two-stage framework involves a discretization process, which prevents differentiable learning and simultaneous optimization. We also pursuit the cost-effective end-to-end solver for further work.

## 5   Conclusion and Outlook

We have proposed HubRouter, a generative global router in VLSI with a two-phase learning scheme: the hub-generation phase and the pin-hub-connection phase. In the generation phase, HubRouter generates hubs instead of routes while in the connection phase, it connects the hubs with RSMT construction. Experiments on real-world and simulated datasets show its effectiveness as well as scalability. When adopting HubRouter, the potential negative impacts should be taken care: 1) The incorrect generation might lead to poor routing results; 2) Generative models are computationally intensive and might consume a lot of resources.

This paper also has some *limitations* for future work: 1) The two-phase scheme is not end-to-end under a joint training scheme; 2) The supervised module for hub generation depends on large amount of training data; 3) Like SOTA generative global routing algorithms [6], the further decrease on overflow needs a reroute process; 4) Finally, it would also be compelling to integrate the learning-based methods of logic synthesis [55], with routing methodologies, including those presented in this work. This would allow for a data-driven approach to the chip design pipeline.

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

# A  Related Works

**Global Routing.** Traditional global routing algorithms typically start with decomposing multi-pin nets into two-pin nets by minimum spanning tree (MST) or rectilinear Steiner tree (RST) [9, 18]. Then, two-pin nets are routed by heuristic-based methods such as pattern routing [24] and maze routing. Finally, a rip-up and reroute [43] process is adopted to refine the final result. Negotiation-based rip-up and reroute algorithms [36, 8] reveal their superiority by considering rerouting history. In addition to classical solutions, recent methods start to resort to learning techniques. RL-based routers with a DQN agent [31] sequentially decide every route direction in a net and aim to generate a better result than A* router, but they are time-consuming and currently often only applicable to small grids, rather than realistic settings. Generative models are recently introduced to generate routes in a net without decomposing it into two-pin nets. [48] uses generative models to speed up global routing at the cost of reduced routability. In PRNet [6], global routing is solved by conditional generative models (specifically CGAN [37]), but it still heavily depends on unscalable post-processing due to some unconnected routes. This paper achieves connectivity by decomposing global routing into a two-phase learning scheme.

**Generative Model.** GAN [11], VAE [26], and DPM [16] are three widely-used deep generative models applied in the generation of different forms of data. Moreover, some works [37, 44, 17] incorporate conditions to make the generation under more fine-grained control, which is important in applications like layout design [5, 56], combinatorial optimization [29], and chip design [6]. Generation tasks also attract the research on the distributions of real-world data [30], which show high potentials. In this paper, we present a novel and more scalable hub generation scheme to address global routing problems.

**RSMT Construction.** Rectilinear Steiner minimum tree (RSMT) gives the shortest possible solution to interconnect pins in a net. This problem belongs to the class of combinatorial problems [12, 57] that are NP-Complete [10] and challenging for machine learning researchers. Since it is computationally intractable to find an optimal solution in general, rectilinear minimum spanning tree (R-MST) [21] is widely used to approximate RSMT with time complexity of $O(n \log n)$ [21], but its length is at most $1.5\times$ of the optimal RSMT [20]. FLUTE [53] designs a look-up table for small nets and decomposes larger nets into smaller nets that can be solved by the table. REST [33] introduces rectilinear edge sequence (RES) to encode RSMT and train an actor-critic neural network by reinforcement learning. Compared to existing RSMT solvers, this paper performs RSMT construction after a hub generation phase, and as such, global routing is completed by connecting the pins and generated hubs.

# B  Algorithm

## B.1  Deep Generative Approaches

We incorporate three prevailing generative models into our generation framework. These approaches give us different perspectives of hub generation:

**GAN [11].** Conditional GAN (CGAN) [37] applies a discriminator $D(\cdot)$ to classify whether the input is real or fake (generated), which minimizes the logistic loss function $-\log D_\phi(p(\mathbf{x}|\mathbf{c})) - \log(1 - D_\phi(p_\theta(\mathbf{x}|\mathbf{c}, \mathbf{z})))$ with regard to $\phi$. It equals maximizing its negative value. Meanwhile, the generated distribution $p_\theta(\mathbf{x}|\mathbf{c}, \mathbf{z})$ is called a generator in GAN, which approximates to $p(\mathbf{x}|\mathbf{c})$ by deceiving the discriminator with regard to $\theta$. Together optimize the above objectives, we have a min-max strategy:

$$\min_\theta \max_\phi \sum_{i=1}^{N} \left[\log D_\phi(\mathbf{x}_i) + \log(1 - D_\phi(p_\theta(\mathbf{x}|\mathbf{c} = \mathbf{c}_i, \mathbf{z} \sim p_\mathbf{z}(\mathbf{z}))))\right]. \tag{4}$$

We use neural networks to learn the discriminator $D_\phi(\cdot)$ and the generator $p_\theta(\mathbf{x}|\mathbf{c}, \mathbf{z})$.

**VAE [26].** Conditional VAE (CVAE) [44] minimizes Kullback-Leibler (KL) divergence to approximate a simulated inference network $q_\phi(\mathbf{z}|\mathbf{x}, \mathbf{c})$ to the intractable posterior distribution $p_\theta(\mathbf{z}|\mathbf{x}, \mathbf{c})$, such that we can apply the stochastic gradient variational Bayes (SGVB) framework to optimize the variational lower bound

$$\log p_\theta(\mathbf{x}|\mathbf{c}) \geq \underbrace{-\text{KL}\left[q_\phi(\mathbf{z}|\mathbf{x}, \mathbf{c}) \| p_\theta(\mathbf{z}|\mathbf{x}, \mathbf{c})\right]}_{\textcircled{1}} + \underbrace{\mathbb{E}_{q_\phi(\mathbf{z}|\mathbf{x}, \mathbf{c})} \log p_\theta(\mathbf{x}|\mathbf{c}, \mathbf{z})}_{\textcircled{2}}. \tag{5}$$

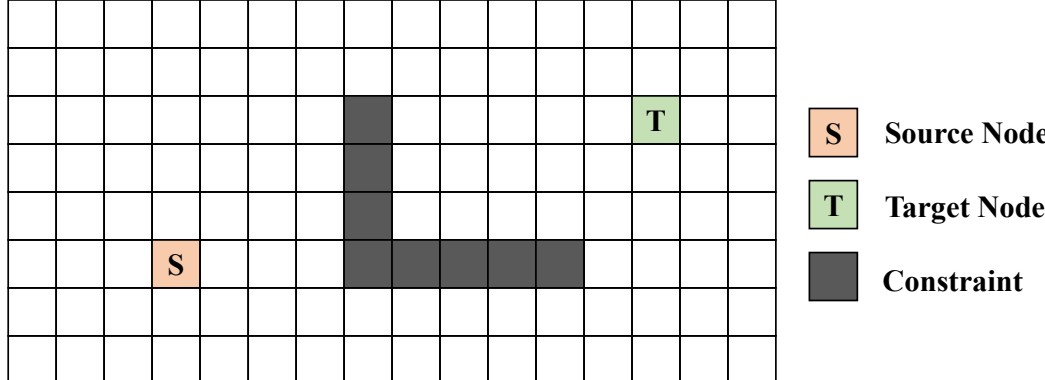

Figure 8: The simplified 2-pin problem, namely, routing the source and target node (pin) under constraints while searching for the shortest wirelength for a single net. The constraints represent the obstacles derived from capacity.

Note that the first term in Eq. 5 is easy to integrate since the KL divergence for Gaussian distribution has closed form, and the second term is often simplified by Monte-Carlo sampling $\mathbb{E}_{q_\phi(\mathbf{z}|\mathbf{x},\mathbf{c})} \log p_\theta(\mathbf{x}|\mathbf{c},\mathbf{z}) \approx \frac{1}{L}\sum_{l=1}^{L} \log p_\theta(\mathbf{x}|\mathbf{c},\mathbf{z}^{(l)})$. So, the objective of minimizing the empirical lower bound can be written as:

$$\min_{\theta,\phi}\left[-\mathrm{KL}\left(q_\phi(\mathbf{z}|\mathbf{x},\mathbf{c})\|p_\theta(\mathbf{z}|\mathbf{c})\right) + \frac{1}{L}\sum_{l=1}^{L}\log p_\theta\left(\mathbf{x}|\mathbf{c},\mathbf{z}^{(l)}\right)\right], \tag{6}$$

where $\mathbf{z}^{(l)} = g_\phi(\mathbf{x},\mathbf{c},\epsilon^{(l)}), \epsilon^{(l)} \sim \mathcal{N}(\mathbf{0},\mathbf{I})$ is a reparameterization trick [26] and $L$ is the number of samples. Neural networks are used to learn the encoder $q_\phi(\mathbf{z}|\mathbf{x},\mathbf{c})$, the decoder $p_\theta(\mathbf{x}|\mathbf{c},\mathbf{z})$, and the conditional prior network $p_\theta(\mathbf{z}|\mathbf{c})$.

**DPM [16].** DPM injects noise into the original data step by step until it turns out to be a known distribution and reverses these steps when sampling. DPM respectively defines these processes as the forward process $q_\theta(\mathbf{x}_t|\mathbf{x}_{t-1})$ and the reverse process $p_\theta(\mathbf{x}_{t-1}|\mathbf{x}_t)$ with $t \in \{1,\cdots,T\}$:

$$q(\mathbf{x}_{1:T}|\mathbf{x}_0) := \prod_{t=1}^{T} q(\mathbf{x}_t|\mathbf{x}_{t-1}), \qquad q(\mathbf{x}_t|\mathbf{x}_{t-1}) = \mathcal{N}(\mathbf{x}_t; \sqrt{1-\beta_t}\mathbf{x}_{t-1}, \beta_t\mathbf{I}),$$

$$\tag{7}$$

$$p_\theta(\mathbf{x}_{0:T}) := p(\mathbf{x}_T)\prod_{t=1}^{T} p_\theta(\mathbf{x}_{t-1}|\mathbf{x}_t), \qquad p_\theta(\mathbf{x}_{t-1}|\mathbf{x}_t) = \mathcal{N}(\mathbf{x}_{t-1}; \mu_\theta(\mathbf{x}_t,t), \sigma_t^2),$$

where $\mathbf{I}$ is an identity matrix, $\{\beta_t\}_{t=1}^{T}$ are fixed variance schedules with ascending values, and $\alpha_t = 1 - \beta_t$. $\{\sigma_t^2\}_{i=1}^{T}$ are also fixed values. Similar to VAE, with training the variational lower bound of the marginal log likelihood and reparameterization, we finally obtain the condition-guided DPM objective:

$$\min_{\theta}\left[\mathbb{E}_{\mathbf{x},\mathbf{c},\epsilon\sim\mathcal{N}(\mathbf{0},\mathbf{I}),t\sim\mathcal{U}(\{t\}_{t=1}^{T})}\left[\|\epsilon - \epsilon_\theta(\mathbf{z}_t,\mathbf{c},t)\|^2\right]\right]. \tag{8}$$

Following classifier-free guidance [17], we initialize $\mathbf{z}_T$ and denoise it by $\hat{\epsilon}_t = (1+w)\cdot\epsilon_\theta(\mathbf{z}_t,\mathbf{c},t) - w\cdot\epsilon_\theta(\mathbf{z}_t,t)$, and accelerate the sampling process with fewer sampling steps by DDIM [45].

## B.2 Training and Sampling

Adam [25] is used to train in both generation and connection phases. Specifically, Alg. 1 and Alg. 2 illustrate the training processes of these two phases, while Alg. 3 shows the sampling process.

---

**Algorithm 1 Training in Hub-generation Phase.**

---

**Input:** Number of iterations $n\_iters$, minibatch size $m$, condition images $\mathbf{c}$, real route images $\mathbf{x}^{(rt)}$, generative model mode $mode \in \{GAN, VAE, DPM\}$, maximum time step $T$ in DPM.

1   **for** $iter$ from 1 to $n\_iters$ **do**

2      Sample minibatch of $m$ condition images $\{\mathbf{c}_1, \cdots, \mathbf{c}_m\}$, real route images $\{\mathbf{x}_1^{(rt)}, \cdots, \mathbf{x}_m^{(rt)}\}$;

3      Compute $\mathbf{x}_i^{(hub)}$ and $\mathbf{x}_i^{(msk)}$ for each $\mathbf{x}_i^{(rt)}$, and compose $\mathbf{x}_i = \{\mathbf{x}_i^{(hub)}, \mathbf{x}_i^{(rt)}, \mathbf{x}_i^{(msk)}\}$ ($i = 1, \cdots, m$);

4      **if** $mode = GAN$ **then**

5         Sample minibatch of $m$ noise images $\{\mathbf{z}_1, \cdots, \mathbf{z}_m\}$, where $\mathbf{z}_i \sim \mathcal{N}(\mathbf{0}, \mathbf{I})$ ($i = 1, \cdots, m$);

6         Ascend the stochastic gradient of the discriminator in Eq. 4:
$$\nabla_\phi \left[ \tfrac{1}{m} \sum_{i=1}^m \left[ \log D_\phi(\mathbf{x}_i) + \log(1 - D_\phi(p_\theta(\mathbf{x}|\mathbf{c} = \mathbf{c}_i, \mathbf{z} = \mathbf{z}_i))) \right] \right];$$

7         Sample minibatch of $m$ noise images $\{\mathbf{z}_1, \cdots, \mathbf{z}_m\}$;

8         Descend the stochastic gradient of the generator in Eq. 4:
$$\nabla_\theta \left[ \tfrac{1}{m} \sum_{i=1}^m \left[ \log(1 - D_\phi(p_\theta(\mathbf{x}|\mathbf{c} = \mathbf{c}_i, \mathbf{z} = \mathbf{z}_i))) \right] \right];$$

9      **else if** $mode = VAE$ **then**

10        Sample minibatch of $m$ noise images $\{\epsilon_1, \cdots, \epsilon_m\}$, where $\epsilon_i \sim \mathcal{N}(\mathbf{0}, \mathbf{I})$ ($i = 1, \cdots, m$);

11        Compute $\{\mathbf{z}_1, \cdots, \mathbf{z}_m\}$ using the reparameterization trick [26];

12        Descend the stochastic gradient of Eq. 6:
$$\nabla_{\theta,\phi} \left[ \tfrac{1}{m} \sum_{i=1}^m \left( - \mathrm{KL}\left(q_\phi(\mathbf{z}|\mathbf{x} = \mathbf{x}_i, \mathbf{c} = \mathbf{c}_i) \| p_\theta(\mathbf{z}|\mathbf{c} = \mathbf{c}_i)\right) + \log p_\theta\left(\mathbf{x}|\mathbf{c} = \mathbf{c}_i, \mathbf{z} = \mathbf{z}_i\right) \right) \right];$$

13                                                `// Let L = 1.`

14      **else if** $mode = DPM$ **then**

15        Sample minibatch of $m$ noise images $\{\epsilon_1, \cdots, \epsilon_m\}$, where $\epsilon_i \sim \mathcal{N}(\mathbf{0}, \mathbf{I})$ ($i = 1, \cdots, m$);

16        Sample minibatch of $m$ time steps $\{t_1, \cdots, t_m\}$, where $t_i \sim \mathcal{U}(\{1, \cdots, T\})$ ($i = 1, \cdots, m$);

17        Descend the stochastic gradient of Eq. 8:
$$\nabla_\theta \left[ \mathbb{E}_{\mathbf{x}, \mathbf{c}, \epsilon \sim \mathcal{N}(\mathbf{0}, \mathbf{I}), t \sim \mathcal{U}(\{t\}_{t=1}^T)} \left[ \| \epsilon - \epsilon_\theta(\mathbf{z}_t, \mathbf{c}, t) \|^2 \right] \right];$$

**Output:** Trained models with parameters $\theta$ and $\phi$.

---

**Algorithm 2 Training in Pin-hub-connection Phase.**

---

**Input:** Number of iterations $n\_iters$, minibatch size $B$.

18   **for** $iter$ from 1 to $n\_iters$ **do**

19      Sample $B$ random point sets $\bar{V} = \{V_1, \cdots, V_B\}$.

20      Construct valid RES sets $R^{(V_i)}$ for each point set $V_i$;

21      Sample a single RES $r_i \sim R^{(V_i)}$ for each point set $V_i$;

22      Linearly evaluate the actual length $b(V_i, r_i)$;

23      Descend the gradient in the actor network according to Eq. 2 using reinforcement algorithm [52]:
$$\tfrac{1}{B} \sum_{i=1}^B [b_\zeta(V_i) - b(V_i, r_i)] \nabla_\xi [p_\xi(r_i)];$$

24      Descend the gradient in the critic network according to Eq. 3:
$$\nabla_\zeta \left[ \tfrac{1}{B} \sum_{i=1}^B \| b_\zeta(V_i) - b(V_i, r_i) \|^2 \right].$$

**Output:** The trained model with parameters $\xi$ and $\zeta$.

---

## C   Definition and Theorem

### C.1   2-pin Problem

Fig. 8 gives an example of the '2-pin' problem, which is the simplest version of global routing, but it turns out to be an NP-Complete problem [27, 31].

### C.2   Bonus of RSMT Reconstruction

**Lemma 1.** *Rectilinear Steiner points are hubs.*

**Algorithm 3 Sampling Routes.**

---

**Input:** Condition image $\mathbf{c}_0$, generative model mode $mode \in \{GAN, VAE, DPM\}$, trained generative models and the RSMT connection model, DDIM [45] timesteps $\{\hat{t}_1, \cdots, \hat{t}_d\}$, guide weight $w$ in DPM.

25   // Hub-generation Phase.
26   **if** $mode = GAN$ **then**
27     |   Sample a noise sample $\mathbf{z}_0 \sim \mathcal{N}(\mathbf{0}, \mathbf{I})$;
28     |   Generate $\mathbf{x}_0 = p_\theta(\mathbf{x}|\mathbf{c} = \mathbf{c}_0, \mathbf{z} = \mathbf{z}_0)$;
29   **else if** $mode = VAE$ **then**
30     |   Sample a noise sample $\mathbf{z}_0 \sim \mathcal{N}(\mathbf{0}, \mathbf{I})$;
31     |   Generate $\mathbf{x}_0 = p_\theta(\mathbf{x}|\mathbf{c} = \mathbf{c}_0, \mathbf{z} = \mathbf{z}_0)$;
32   **else if** $mode = DPM$ **then**
33     |   Sample a noise sample $\mathbf{x}_{\hat{t}_d} = \epsilon_{\hat{t}_d} \sim \mathcal{N}(\mathbf{0}, \mathbf{I})$;
34     |   **for** $\hat{t}_j$ from $\hat{t}_d$ to $\hat{t}_1$ **do**
35     |     |   Compute previous $\alpha_{\hat{t}_{j-1}}$ and current $\alpha_{\hat{t}_j}$;
36     |     |   Predict the noise: $\hat{\epsilon}_{\hat{t}_j} = (1 + w) \cdot \epsilon_\theta(\mathbf{x}_{\hat{t}_j}, \mathbf{c}_{\hat{t}_j}, \hat{t}_j) - w \cdot \epsilon_\theta(\mathbf{x}_{\hat{t}_j}, \hat{t}_j)$;
37     |     |   Predict $\mathbf{x}_0$: $p_{\mathbf{x}_0} \leftarrow (\mathbf{z}_{\hat{t}_d} - \sqrt{1 - \alpha_{\hat{t}_j}} \hat{\epsilon}_{\hat{t}_j}) / \sqrt{\alpha_{\hat{t}_j}}$;
38     |     |   Compute variance: $\sigma_{\hat{t}_j}(\eta) = \eta \sqrt{(1 - \alpha_{\hat{t}_{j-1}})/(1 - \alpha_{\hat{t}_j})(1 - \alpha_{\hat{t}_j}/\alpha_{\hat{t}_{j-1}})}$;
39     |     |                     // We directly set $\eta = 0$ here.
40     |     |   Compute the direction pointing to $\mathbf{x}_{\hat{t}_j}$: $d_{\mathbf{x}_{\hat{t}_j}} = \sqrt{1 - \alpha_{\hat{t}_{j-1}} - \sigma_{\hat{t}_j}^2} \cdot \hat{\epsilon}_{\hat{t}_j}$;
41     |     |   Compute $\mathbf{x}_{\hat{t}_{j-1}}$: $\mathbf{x}_{\hat{t}_{j-1}} = \sqrt{\alpha_{\hat{t}_{j-1}}} \cdot p_{\mathbf{x}_0} + d_{\mathbf{x}_{\hat{t}_j}} + \sigma_{\hat{t}_j}(\eta) \cdot \epsilon_{\hat{t}_j}$, where $\epsilon_{\hat{t}_j} \sim \mathcal{N}(\mathbf{0}, \mathbf{I})$.
42     |   Finally, we have $\mathbf{x}_0$;
43     |   // Denote $\mathbf{x} = \mathbf{x}_0$, and $\mathbf{x}_{ij}$ as its element in $i$-th row and $j$-th column.
44   Compute the binarized strip mask $\tilde{\mathbf{x}}^{(msk)}$ according to Eq. 1, where each $\tilde{\mathbf{x}}_{ij}^{(msk)}$ is computed as:

$$\tilde{\mathbf{x}}_{ij}^{(msk)} = \mathbb{I}\left[ \max\left( \frac{1}{m} \sum_{i=1}^{m} \mathbf{x}_{ij}^{(msk)}, \frac{1}{n} \sum_{j=1}^{n} \mathbf{x}_{ij}^{(msk)} \right) > \frac{1}{2} \right];$$

45                                  // Note that $\mathbf{x} = \{\mathbf{x}^{(hub)}, \mathbf{x}^{(rt)}, \mathbf{x}^{(msk)}\}$.
46   Compute the masked hubs: $\hat{\mathbf{x}}_{ij}^{(hub)} = \mathbf{x}_{ij}^{(hub)} \times \tilde{\mathbf{x}}_{ij}^{(msk)}$;
47   Obtain the union of hubs and pins $V$;
48   // Pin-hub-connection Phase.
49   Obtain the RES $r$ according to the maximum $p_\xi(r_i)$, where $r_i \sim R^{(V)}$;
50   Compare 8 transformations introduced in Sec. 3.2 and pick the optimal route with the shortest length.
    **Output:** The generated route and its length.

---

**Proof** According to Hanan theory [13], the rectilinear Steiner points can be restricted to the Hanan grid. The definition of hubs in this paper implies that all the routed points on the intersection of Hanan grid are hubs, so rectilinear Steiner points must be hubs.    □

**Remark 1.** *[Reconstruction Bonus] Suppose a set of hubs in an RSMT route is correctly generated, then its RSMT reconstruction can be performed in $O(n \log n)$ time complexity.*

**Proof** According to Lemma 1, since all the rectilinear Steiner points are hubs, there is no need to introduce extra rectilinear Steiner points when generating an R-MST towards all the hubs. So, the RSMT problem is degraded to an R-MST problem, which has been proved to be addressed in $O(n \log n)$ time complexity [21].    □

## D   Experimental Protocols

### D.1   Dataset.

Real-world Datasets ISPD-07 [38] and ISPD-98 [1], and the simulated dataset constructed by [31] (we call 'DRL') are employed in this work.

Table 6: Summary of the test dataset. We respectively show the scale size, vertical/horizontal capacity, number of nets, and average/maximum number of pins for each net.

| CASE | IBM01 | IBM02 | IBM03 | IBM04 | IBM05 | IBM06 | DRL-8 | DRL-16 |
|---|---|---|---|---|---|---|---|---|
| SIZE | 64×64 | 80×64 | 80×64 | 96×64 | 128×64 | 128×64 | 8×8 | 16×16 |
| CAP. (V / H) | 24 / 28 | 44 / 68 | 40 / 60 | 40 / 46 | 84 / 126 | 40 / 66 | 4 / 4 | 6 / 6 |
| #NETS | 11507 | 18429 | 21621 | 26163 | 27777 | 33354 | 20 | 30 |
| AVG. #PINS | 4.31 | 4.88 | 4.10 | 3.86 | 5.25 | 4.21 | 5.25 | 6.12 |
| MAX #PINS | 42 | 134 | 55 | 46 | 17 | 35 | 8 | 10 |

In line with [6], we construct training datasets by adopting [34] to route on ISPD-07 benchmarks and we declare the details unstated in [6]. We initialize the capacity given by the benchmarks and sequentially route the nets using the results of [34]. Each time the capacity is updated, a condition image, consisting of the current capacity and the positions of pins to be routed in the next net, is generated. Meanwhile, a ground-truth route image is generated and saved correspondingly. By clipping them to the same scale $64 \times 64$ (if possible) randomly, the image will be saved to different directories, i.e., *Route-small-4*, *Route-small*, *Route-large-4*, and *Route-large*, according to the number of pins $n_p$ and their HPWL. Specifically,

1) *Route-small-4*: $n_p \leq 4$ and $HPWL \leq 16$;

2) *Route-small*: $n_p > 4$ and $HPWL \leq 16$;

3) *Route-large-4*: $n_p \leq 4$ and $HPWL > 16$;

4) *Route-large*: $n_p > 4$ and $HPWL > 16$.

Examples are partly shown in Fig. 10. This division on the one hand originates according to the observation from [6] that the average pin number is about 4 in most chip cases, as shown in Table 6, and on the other hand, we would like to show the non-connectivity of previous generative global routing approaches under different situations. We choose bigblue4, newblue3, newblue4, and newblue7 (name of the cases in ISPD-07) as the training cases and generate 15K training samples for each case's each category. Thus, we have a total of nearly 240K (some categories cannot generate a maximum of 15K samples). In Table 2, we test the cases of adaptec1, adaptec2, newblue1, newblue2, bigblue1, and bigblue2 in ISPD-07 with a total 10K samples for each category.

Cases in ISPD-98 and DRL are used as test data in Table 3 and Fig. 5; however, their scales are not the same as the input data. So, we clip the ISPD-98 and pad the DRL to ensure scale consistence. Very few (<0.1%) nets in ISPD-98 cannot satisfy the input scale due to their exceeded pin positions. For these special samples, we divide them into multiple samples and then feed them to the model. Note that there might be better solutions to deal with the scale problem, e.g., the input-size-adapting module in [6], but it is not the main focus of this work. Moreover, two simulated dataset DRL-8 and DRL-16, whose scales are respectively $8 \times 8$ and $16 \times 16$, are generated using the tool [1] introduced by [31]. We randomly generate 10 cases for DRL-8 and DRL-16, respectively. More information about the test dataset is detailed in Table 6.

## D.2    Baselines.

The baselines referred in Table 3 are introduced as follows:

1) *Labyrinth* [24], a classical routing algorithm that discusses how the concept of pattern routing can be used to guide the router to a solution that minimizes interconnect delay without damaging the routability of the circuit;

2) *BoxRouter* [8], a classical global router that first employs a pre-routing strategy to identify the most congest area and then performs box expansion and progressive integer linear programming(ILP);

3) *DQN* [31], a reinforcement learning global routing algorithm with A* router burn-in memory and conjoint optimization;

---

[1] https://github.com/haiguanl/DQN_GlobalRouting

4) *PRNet* [6], a joint learning framework that together addresses the placement and routing problems, where the routing is based on generative approches.

In the RSMT construction experiment in Sec. 4.4, the referred baselines are:

1) *GeoSteiner* [51], the optimal RSMT construction solver;

2) *R-MST*, an efficient implementation of R-MST construction with time complexity of $O(n \log n)$;

3) *BGA* [23], a practical $O(n \log^2 n)$ sub-optimal solution of RSMT construction with heuristics.

4) *FLUTE* [53], a fast and accurate RSMT construction approach by using a look-up table. Note that it can reach the optimal solution for nets up to 9 degrees.

5) *REST* [33], a state-of-the-art RL-based approach that uses an actor-critic network to predict the Rectilinear edge sequence (RES) for RSMT construction.

### D.3 Model Setting.

In the hub-generation phase, VAE employs a CNN [48] structure while GAN uses a ResNet [14, 22] structure, which is in line with [6], but we make some revisions. For instance, [6] utilizes a bi-discriminator [54] to inject the correctness constraint to the objectives, but our proposed HubRouter no longer requires it within the two-phase learning scheme. In addition, the model structure of DPM follows [17] due to the condition guidance and uses a U-Net [41] structure, but we make it lightweight to achieve more rapid sampling time. Fig. 9 details the components of the generative models in this work. In the pin-hub-connection phase, we follow REST [33] to use a multi-head attention encoder [49] in its auto-encoder framework.

## E    Experimental Analysis

### E.1    Hyperparameters

We search hyperparameters on the validation dataset with the learning rate in [0.001, 0.0001] and reduce the learning rates by 0.96 after every 10 epochs until the validation loss no longer decreases for over 20 epochs or the number of epochs reaches a maximum of 200. In the hub-generation phase, we optionally choose three generative models, including GAN, VAE, and DPM. For GAN and VAE, we use similar structures and search the number of ResNet blocks in [3, 6, 9] and the number of downsampling/upsampling layers in [2, 3, 4]. For DPM, we search the number of DDIM steps in [25, 50, 75], the guide weight $w$ in [0.0, 1.5, 2.0], and the maximum timestep in [500, 1000]. The best models are chosen by the optimal wirelength rate while keeping the generation time tolerable. Each experiment is trained with a batch size of 64 and the optimizer Adam [25] with the decay rate of first(second)-order moment estimation $0.5(0.999)$ and the L2 penalty coefficient 0.01. In the experiments in Sec. 4, the number of ResNet blocks and the number of downsampling/upsampling layers for GAN and VAE are respectively set as 9 and 2. For DPM, we set the number of DDIM steps as 50, the guide weight $w$ as 1.5, and the maximum timestep as 500. In the pin-hub-connection, we follow REST [1] [33] and employ the model with adjusted parameters.

### E.2    Correspondence between Hub and Route

The main idea of the two-phase learning scheme originates from our observation and curiosity - whether the hub and the route have high correspondence. If so, we could conclude that the optimal final routes would approximately be the same for generating hubs or generating routes directly. According to the definition in Def. 1, it is apparent that each route can uniquely determine its hubs, but regretfully the opposite is not true. That is to say, when given a group of hubs, we cannot decide uniquely what their original route is; however, we empirically find the correspondence between hubs and routes is high. To show this, we generate the hubs according to the definition for each net in ISPD-07 and then use these generated hubs and pins to construct RSMT. As shown in Table 7, the correspondence rates are respectively 99.7% and 98.18% for *Route-small-4* and *Route-large-4* and importantly these two categories occupy large proportions (totally over 80%) in ISPD-98 and ISPD-07 cases. Though the correspondence rate in *Route-large* is not high, the mean square error

---

[1] https://github.com/cuhk-eda/REST

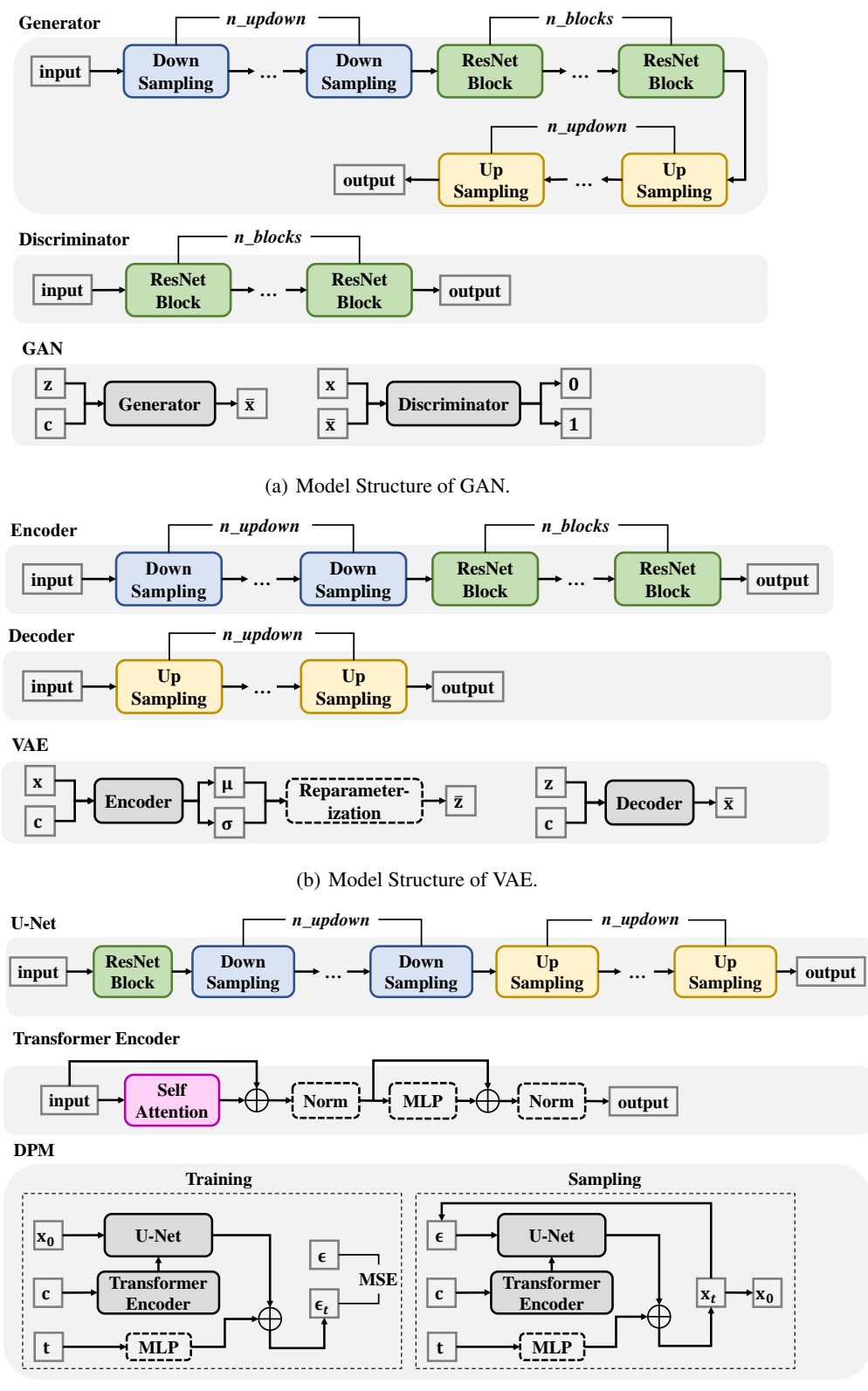

(a) Model Structure of GAN.

(b) Model Structure of VAE.

(c) Model Structure of DPM.

Figure 9: Model Components of a) GAN; b) VAE; c) DPM.

Table 7: Correspondence between hubs and routes. The correspondence rate and mean square error (MSE) are computed on ISPD-07 cases.

| Category | Route-small-4 | Route-small | Route-large-4 | Route-large |
|---|---|---|---|---|
| Correspondence Rate (%) | 99.7 | 80.31 | 98.18 | 48.75 |
| MSE | $< 10^{-5}$ | 0.0004 | 0.0002 | 0.003 |
| Proportion in ISPD-98 (%) | 65.38 | 4.69 | 15.94 | 13.99 |
| Proportion in ISPD-07 (%) | 66.25 | 9.67 | 14.89 | 9.19 |

Table 8: Relative error on ISPD-98 cases. The first line introduces the theoretical lower bound introduced in [7]. Optimal results are in **bold**.

| Model | IBM01 | IBM02 | IBM03 | IBM04 | IBM05 | IBM06 |
|---|---|---|---|---|---|---|
| Lower Bound [7] | 60142 | 165863 | 145678 | 162734 | 409709 | 275868 |
| Labyrinth [22] | 0.262 | 0.589 | 0.693 | 0.551 | 0.181 | 1.093 |
| BoxRouter [7] | 0.059 | 0.107 | 0.03 | 0.105 | **0.015** | 0.077 |
| PRNet [6] | 0.03 | 0.115 | 0.106 | 0.129 | 0.176 | 0.198 |
| HR-GAN | **0.015** | **0.028** | **0.023** | **0.026** | 0.036 | **0.039** |

(MSE) between the generated route and the real one is still extremely low. These observations ensure that the hub generation can be effective for global routing.

## E.3 Relative Error on ISPD-98 cases

To judge the improvement of gaining on the optimal wirelength rather than the absolute value, we further compare the relative error in Table 8, where the relative error is computed as $(WL - b)/b$. Here, $b$ denotes the theoretical lower bound [8]. As shown in this table, the WL promotion of HubRouter is notable compared with the SOTA generative global routing method (PRNet).

## E.4 Generation Time and Connection Time

We present the breakdown of the inference time for each phase on ISPD-98 (IBM01-06) and GRL cases in Table 9. From an overall perspective, the connection phase has comparable time costs for the three generative models. The generation phase, however, dominates the overall time consumption. This indicates that HubRouter achieves faster performance than other generative models by reducing the time spent in the connection phase.

Table 9: Generation and connection time in the experiments on ISPD-98 (IBM01-06) and GRL cases.

| Metric | Model | IBM01 | IBM02 | IBM03 | IBM04 | IBM05 | IBM06 | GRL-8 | GRL-16 |
|---|---|---|---|---|---|---|---|---|---|
| **Generation Time (Sec.)** | HR-VAE | 8.13 | 6.68 | 7.47 | 8.01 | 9.90 | 12.18 | 5.78 | 5.77 |
| | HR-DPM | 1802 | 2841 | 3014 | 3839 | 4329 | 5110 | 36.76 | 49.55 |
| | HR-GAN | 40.12 | 45.36 | 49.02 | 61.33 | 68.47 | 91.50 | 6.04 | 6.26 |
| **Connection Time (Sec.)** | HR-VAE | 1.67 | 3.23 | 2.87 | 5.09 | 5.08 | 5.04 | 0.04 | 0.11 |
| | HR-DPM | 1.80 | 3.51 | 2.88 | 4.99 | 5.04 | 4.98 | 0.04 | 0.11 |
| | HR-GAN | 1.72 | 3.30 | 2.84 | 5.00 | 5.04 | 5.11 | 0.04 | 0.11 |

## E.5 Route Generation Results

Examples of the condition and real route images, as well as the route generation results of *Route-small-4*, *Route-small*, *Route-large-4*, *Route-large* introduced in Appendix D.1 are depicted in Fig. 10.

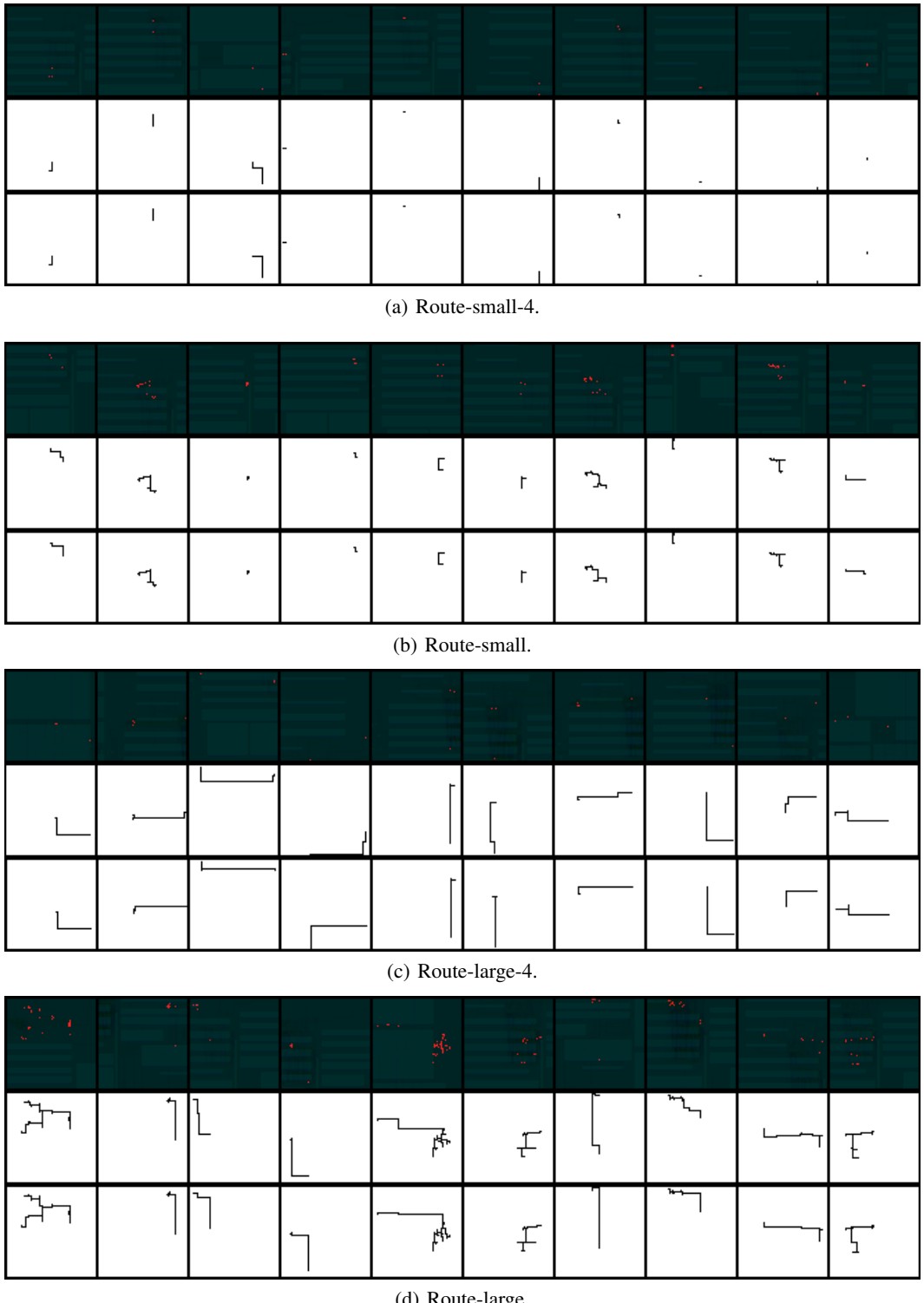

(a) Route-small-4.

(b) Route-small.

(c) Route-large-4.

(d) Route-large.

Figure 10: Condition images (first line), Real route images (second line), and the route images generated by our propose two-phase learning scheme (third line), randomly sampled in a) *Route-small-4*; b) *Route-small*; c) *Route-large-4*; d) *Route-large*.

