# OpenReview forum: "HubRouter: Learning Global Routing via Hub Generation and Pin-hub Connection"
_NeurIPS.cc/2023/Conference — NeurIPS 2023 poster_

### Official Review · Reviewer_GN8b · 2023-07-03

**Soundness:** 3 good
**Presentation:** 4 excellent
**Contribution:** 3 good
**Rating:** 6
**Confidence:** 3

**Summary:**

This paper proposes a two-phase learning framework called HubRouter for global routing in chip design. Different from previous works that directly generate routes from chip images, which potentially cause inconnectivity, this paper proposes to generate hubs representing tiles in the first phase and then construct RMST with hubs by an actor-critic model. The experimental results show the effectiveness in terms of higher correctness, and shorter WL compared with SOTA model.

**Strengths:**


1. the paper firstly proposes hubs for the global routing task in chip design, transferring the pin-pin problem into hub-pin problems, which can passively avoid the in-connectivity problem when regarding the routing design process as image generation.

2. When dealing with large chips, HubRouter is the best performer among SOTAs regarding both correctness and wire length.

3. The proposed model can have good scalability with GAN or VAE as their generative model for the hub generation phase while maintaining the property of making wire length short and overflow less.

4. In the hub-generation phase, besides hubs, it also generates routes and masks to avoid noises brought by the generative model itself since the noises can greatly impact the results. Especially, the stripe mask can be greatly helpful for complicated cases.


**Weaknesses:**

1.	The overflow still exists for those generated global routing even though the proposed model reduces overflow better than PRNet.
2.	Even though HubRouter can get a good performance on replicating the known facts, it does not discuss the quality of the generated routes regarding congestion, and it possibly cannot generate novel routes since it has limited knowledge of routing design.

**Questions:**

Q1. It is unclear how the stripe mask is generated for chips.

Q2. Why not define pixels as hubs that only connect two neighbors on the same row or columns, i.e., $r_{(i-1)j}+r_{(i+1)j}+r_{i(j-1)}+r_{i(j+1)}=2$. Without points between hubs defined in the paper, how does the second phase generate the routes in the right direction?


**Limitations:**

The authors discuss the limitations of the lack of training data about hub generation, and the model is not an end-to-end model. Still, it also should discuss how the ground truth comes and the potential impact of adopting such methods when generating it.

---

> ### Author Rebuttal · Authors · 2023-08-09
>
> Thank you for your time and valuable feedback, as well as your positive comments and interest in our paper. According to your constructive comments, we make some replies to the questions.
>
> > **W1:The overflow still exists for those generated global routing.**
>
> Yes, overflow still exists but HubRouter achieves better overflow performance compared with the SOTA generative global routing method (PRNet). Specifically, as shown in Figure 5, HubRouter (GAN) can achieve an average overflow reduction of about 40% compared to PRNet on the ISPD-98 benchmarks.
>
> > **W2:It does not discuss the quality of the generated routes regarding congestion, and it possibly cannot generate novel routes since it has limited knowledge of routing design.**
>
> The metric of overflow can reflect the congestion of the generated routes and we can see that HubRouter surpasses the SOTA generative global routing method (PRNet) on overflow. Since the routes are learned from the training datasets, HubRouter cannot generate novel routes when given new design rules, but it can indirectly learn the latent design knowledge from the training datasets.
>
> > **Q1:It is unclear how the stripe mask is generated for chips.**
>
> We define the stripe mask in lines 171-172 and one can generate the stripe mask according to the definition when given a route. The corresponding sketch is shown in Figure 4(b), where gray stripes represent the stripe mask.
>
> > **Q2:Why not define pixels as hubs that only connect two neighbors on the same row or columns? Without points between hubs defined in the paper, how does the second phase generate the routes in the right direction?**
>
> HubRouter's aim is not to generate all pixels in a route but to generate the key points in the first phase, as such in the second phase, HubRouter can connect these key points with the pins. If pixels that only connect two neighbors on the same row or column are also regarded as hubs, then the hubs will be the same as all pixels in a route, which cannot correspond to our original purpose.
>
> In the second phase, any two points can be connected under the RSMT connection perspective, and moreover, with the guidance of correctly generated key points (hubs), the routes can be generated as expected.
>
> > **Limitations: It also should discuss how the ground truth comes and the potential impact of adopting such methods when generating it.**
>
> Due to the limited length of the main text, we discussed how the ground truth comes in Appendix D.1.
>
> The potential negative impacts include: 1) the incorrect generation might lead to poor routing results; 2) Generative models require large amounts of computing resources, which might cause a waste of resources. We will incorporate these impacts in the revised version.
>
> Please let us know if you had further questions.

---

> > ### Comment · Reviewer_GN8b · 2023-08-18
> > **Thanks for authors' response**
> >
> > Thanks for the clarifications. Most of my concerns are addressed. After reading the additional materials and discussions, I think this paper shows good potential to assist chip design in terms of WL, correctness rate, and efficiency. However, as a generative model, it mainly generates routes within the training set knowledge. I would like to keep my current score.

---

> > > ### Author Response · Authors · 2023-08-19
> > > **Thanks**
> > >
> > > We appreciate your positive feedback and your support for our work. We are glad that our clarifications and additional materials have addressed most of your concerns. Thank you for taking the time to review our work.

---

### Official Review · Reviewer_Vi96 · 2023-07-03

**Soundness:** 3 good
**Presentation:** 3 good
**Contribution:** 3 good
**Rating:** 7
**Confidence:** 4

**Summary:**

The paper focuses on the generative global routing tasks and mainly ensures the connectivity of generated routes via a two-stage framework. In the first phase, the approach involves a typical generative task, which exploits multi-task learning to promote the generation quality and utilizes a trick called stripe mask to decrease some redundant noise points. In the second phase, the work is formulated as an RSMT construction problem and addresses this problem by REST. The authors show that with correctly generated hubs, the RSMT construction can be solved with less time.

**Strengths:**

+ The structure that generates hubs first and then connects them with pins is novel and interesting, and is reasonable to guarantee the connectivity of routes in the global routing.
+ The motivation is clear, and the authors show in Table 2 that the so-called `unconnectivity’ caused by existing generative global routing algorithms (PRNet) is severe, but I would have preferred the authors to also refer to it in the introduction part to strengthen the motivation.
+ The proposed approach performs better than other generative global routing algorithms in several metrics, especially the connectness rate is 100% and the running time of HubRouter (GAN) is much less than PRNet (GAN).
+ The authors also give some applications other than global routing to show the generality of the proposed approach.

**Weaknesses:**

- The approach is clearly divided into two different phases, but the running time shown in the experiment seems to be combined. The authors could show both generation time and connection time to display the time overhead in either phase.
- Some possible typos: The $r_{(n+1)j}$ should be $r_{(m+1)j}$ in Definition 1.

**Questions:**

- Whether the nets in ISPD98 (Table 3) are routed sequentially? If so, what’s the performance of HubRouter when nets are concurrently routed?

---

> ### Author Rebuttal · Authors · 2023-08-09
>
> Thank you for your time and valuable feedback, as well as your positive comments and interest in our paper. According to your constructive comments, we make some replies to the questions.
>
> > **W1: Displaying time overhead in either phase.**
>
> The time overhead of the two phases is shown in Table 2 in the rebuttal PDF. As can be seen, the connection phase has comparable time costs for the three generative models, which is reasonable. The generation phase, however, dominates the overall time consumption.
>
> > **W2: Some possible typos.**
>
> Thanks again for your meticulous review and we will correct typos in the revised version.
>
> > **Q1: Whether the nets in ISPD98 (Table 3) are routed sequentially? If so, what’s the performance of HubRouter when nets are concurrently routed?**
>
> Yes, the nets are routed sequentially. Such a way of routing can maintain the performance of WL and overflow, but is slower than concurrent routing. The results of concurrent routing are shown in Table 3 and Figure 1 in the rebuttal PDF. The batch size of concurrent routing is set uniformly to 20. As can be seen, the result of concurrent routing is a little worse than the one of sequential routing, but it wins in speed.
>
> Please let us know if you had further questions.

---

> > ### Comment · Reviewer_Vi96 · 2023-08-17
> > **Thanks**
> >
> > Thanks for your rebuttal. I have no more questions and keep my positive score.

---

> > > ### Author Response · Authors · 2023-08-18
> > > **Thanks**
> > >
> > > Thank you for taking the time to read our rebuttal and maintaining your positive score. We greatly appreciate your support of our work.

---

### Official Review · Reviewer_2ZCa · 2023-07-06

**Soundness:** 2 fair
**Presentation:** 2 fair
**Contribution:** 2 fair
**Rating:** 4
**Confidence:** 3

**Summary:**

This paper investigates the issue of global routing in VLSI systems and introduces HubRouter, a method that initially generates hubs and subsequently connects them to pins. In the first phase, the authors explored different generative models. In the second phase, the authors employs an actor-critic model to generate a final routing.


**Strengths:**

1. The paper is well-written and easy to follow.
2. The authors conducted many experiments under different baselines.


**Weaknesses:**

1. The two phases are independent, meaning that the feedback from the second phase does not influence the hub generation, potentially leading to suboptimal results.

2. The authors tried three generative models (VAE, DPM and GAN) in the paper. But the paper lacks clarity on which model should be used in specific situations.

3. The performance improvement achieved is marginal.

**Questions:**

1. In the first phase, the authors introduced three objectives: the hub, route, and mask, with the latter two serving as auxiliary designs. Given that the goal of the route is similar to that of the second phase, it would be worth considering why the authors did not design the two phases as a loop, utilizing the second evaluation result as feedback for the first generation. Furthermore, it would be helpful to understand the differences between the results of the route and the second phase.
Besides, the design of mask is rather hand-craft. Why the threshold is 1/2 in each row and each colum? Why the authors only evaluate the row and column, not the density of a specific rectangle area?

2. In the experiments, it seems that the three generative models (VAE, DPM, and GAN) have their own advantages and disadvantages. For instance, GAN achieves the best WLs on six datasets, DPM has the best WL on the remaining two datasets, and VAE is the fastest. How should one select the model when presented with a new dataset?

3. Considering the absolute value of WL, which is the primary evaluation metric, could the authors explain the real-world benefits of improving WL by 1%?


**Limitations:**

Overall, I believe that the integration of the generative model and reinforcement learning algorithm could be more elegant, potentially leading to further performance improvements.

---

> ### Author Rebuttal · Authors · 2023-08-09
>
> Thank you for your time and valuable feedback. According to your constructive comments, we make some replies to the questions.
>
> > **W1: The two phases are independent, potentially leading to suboptimal result./Q1: Why the authors did not design the two phases as a loop?**
>
> We do not design an end-to-end model mainly because: 1) Some calculations in the two phases are not continuous, which make it non-trivial to incorporate the gradients computing in the backward propagation; 2) The performance of the end-to-end SOTA (PRNet) method is even worse than HubRouter, probabaly due to the complexity of the problem.
>
> We detailedly highlight the reasons in the first question in the global response. Please refer to it for a more complete explanation.
>
> > **W2: The paper lacks clarity on which model should be used in specific situations./Q2: How should one select the model when presented with a new dataset?**
>
> Thanks for your suggestions. We do have tried three generative models in our paper's experiment: VAE, GAN, and DPM.
>
> Here are our added conclusive remarks:
>
> We empirically show that they have different behaviors.
>
> When given a new dataset, we argue that the choice of generative algorithms is not based on the characteristic of the dataset, but the industrial needs. Specifically, if one needs high quality, GAN is the best. If speed is highly required, VAE is the best. If one has abundant resources, DPM can be adopted.
>
> > **W3: The performance improvement achieved is marginal./Q3: Could the authors explain the real-world benefits of improving WL by 1%?**
>
> We respectively disagree that the performance improvement is marginal.
>
> First, the 1% improvement of WL is significant. Actually, the absolute value of WL is a common metric in global routing since it directly shows the performance in each case. Per your comments, it seems that improvement in the absolute value of WL is marginal. However, there exists the theoretical lower bound of this task [1] and it is unfair to judge the improvement on its absolute value. To show this, we compare the relative error in Table 1 in the rebuttal PDF, where the relative error is computed as $(WL-b)/b$. Here, $b$ denotes the theoretical lower bound. As shown in this table, the WL promotion of HubRouter is notable compared with the SOTA generative global routing method (PRNet).
>
> Second, the ensurance of routing connectivity is crucial in global routing. As can be seen in Table 2, PRNet particularly suffers from severe unconnectivity. Especially for the complex case ‘Route-large’, PRNet only maintains a 4% correctness rate, which means that 96% of samples require the time-consuming post-process. On the contrary, HubRouter can still maintain the connectivity (promote from 4% to 100% on ‘Route-large’).
>
> Third, the promotion of inference time is significant and this is a metric vital to industrial applications. As shown in Table 3, HubRouter is on average 12x faster than SOTA generative global routing method (PRNet) on ISPD-98 cases because the time-consuming post-processing to achieve connectivity is not required for HubRouter.
>
> In conclusion, HubRouter's contribution is absolutely not just improving WL by 1%. We argue that the significant improvement on WL, correctness rate, and inference time can substantially boost the efficiency and quality of global routing, which is vital to the chip design.
>
> > **Q1: It would be helpful to understand the differences between the results of the route and the second phase.**
>
> The first phase can generate the route but it suffers from the unconnectivity like PRNet. So, the routes generated in the first phase are only auxiliarily used to guide the hub generation by multi-task learning. In the second phase, with a connection process, the route can be ensured to be connected.
>
> > **Q1: the design of mask is rather hand-craft. Why the threshold is 1/2 in each row and each column?**
>
> We use the majority voting algorithm [2] to determine whether a row/column is a stripe mask based on the pixel classification. This algorithm is commonly applied in machine learning, such as ensemble learning [3]. The criterion is that if more than half of the pixels are classified as 1, the row/column is considered a stripe mask. Although this is a theoretically sound threshold, we acknowledge that it may not be optimal in practice. However, we observe that the pixel ratio of any row/column that is a stripe mask is higher than 0.9, and lower than 0.1 for those that are not. This suggests that the model has learned the feature of the stripe mask well. Therefore, changing the threshold to 0.4 or 0.6 would not affect the results.
>
>
>
> > **Q1: Why the authors only evaluate the row and column, not the density of a specific rectangle area?**
>
> We have tested several tricks to promote the generation quality and finally chose the stripe mask. Corresponding reasons are given at the end of Section 3.1.
>
>
> Please also refer to the global response and let us know if you had further questions.
>
> Reference:
>
> [1] BoxRouter 2.0: Architecture and implementation of a hybrid and robust global router, ICCAD 2007
>
> [2] A theoretical analysis of the application of majority voting to pattern recognition, ICPR 2014
>
> [3] A weighted majority voting ensemble approach for classification, ICSE 2019

---

### Official Review · Reviewer_Co1X · 2023-07-31

**Soundness:** 2 fair
**Presentation:** 3 good
**Contribution:** 2 fair
**Rating:** 5
**Confidence:** 2

**Summary:**

This paper presents a new two-phase learning approach, called HubRouter, to address the issue of unconnectivity in the generated routes of global routing (GR) tasks in VLSI systems. It has two steps. Firstly, a deep generative model generates a 'hub,' which acts as a key point in the route; then secondly, HubRouter involves an actor-critic model-based RSMT construction module to connect the hubs. This shift from a pin-pin connection to a hub-pin connection method solves the unconnectivity problem in generative approaches. The HubRouter system ensures all generated routes are connected, eliminating the need for time-consuming post-processing. Experimental results show that HubRouter outperforms other state-of-the-art generative global routing models in wirelength, overflow, and time efficiency. It also finds application in RSMT construction and interactive path replanning, demonstrating its versatility.

**Strengths:**

The paper introduces a novel approach, HubRouter, to global routing, proposing a hub generation and hub-pin connection scheme that effectively addresses the challenge of route unconnectivity.

The experimental results show HubRouter outperforming existing generative global routing models in terms of wirelength, overflow, and time efficiency.

The approach is very general. The authors show that the approach can also be applied to RSMT construction and interactive path replanning, demonstrating the versatility of their method.


**Weaknesses:**

The challenge of connectivity problem is unclear. Specifically, it introduces a novel approach to tackling the unconnectivity problem in global routing, fails to establish the significance and relevance of this problem adequately.  It is helpful to elaborate more about the difficulty and significance of connectivity problem.

The effectiveness of the second phase is dependent on the quality of the hubs generated in the first phase. If the generative models do not create effective hubs, the entire approach could be compromised.

**Questions:**

Overall, this paper solves the unconnectivity problem in the generated routes of global routing (GR) tasks in VLSI systems. The experimental results are promising. The efficiency of the proposed model is verified both theoretically and experimentally. However, the reviewer has two concerns:

1. The challenge of connectivity problem is unclear. It is helpful to elaborate more about the difficulty and significance of connectivity problem.

2. The effectiveness of the second phase is dependent on the quality of the hubs generated in the first phase. If the generative models do not create effective hubs, the entire approach could be compromised.

---

> ### Author Rebuttal · Authors · 2023-08-09
>
> Thank you for your time and valuable feedback. Our replies to the questions are as follows.
>
> > **W1/Q1: The background on the challenge of connectivity is unclear.**
>
> Thanks for your suggestion.
>
> Existing generative global routing methods adopt an end-to-end model and suffer from the connectivity problem. It requires a maze routing post-process procedure to connect all routes when they are not connected, which is rather time-consuming and can cause extra overflow. Our proposed approach ensures connectivity via a novel two-phase routing procedure and reduces time overhead significantly compared to the SOTA PRNet (Cheng et al, NeurIPS 2022).
>
> In particular, Table 2 shows that PRNet suffers from severe unconnectivity. Especially for the complex case 'Route-large', PRNet only maintains a 4% correctness rate, which means that 96% of samples require the time-consuming post-process. This leads to the phenomenon that though the generation time of PRNet in Table 2 is about 2x of HubRouter, the total time (generation + post-process) in Table 3 is about 12x of HubRouter on average.
>
> We fully agree that it would be more helpful if the statement of the difficulty and significance of the connectivity problem were put forward in the introduction part. We will make this revision in the new version.
>
> > **W2/Q2: The effectiveness of the second phase is dependent on the quality of hubs generated in the first phase, which may lead to suboptimal solutions.**
>
> We admit that the two-phase routing model may lead to suboptimal solutions. However, global routing is such a complicated task that previous methods like PRNet, which is an end-to-end router, lead to an even worse solution compared to our HubRouter.
>
> We also inject the stripe mask into the model to discard the wrongly generated hubs to improve the robustness. Moreover, even if some of the correct hubs are not generated, the second phase can also have the chance to connect the correct route through a rectilinear polyline.
>
> Please also refer to the global response and let us know if you had further questions.

---

> > ### Comment · Reviewer_Co1X · 2023-08-20
> >
> > Thank you for these clarifications. I have no further concerns.

---

### Author Rebuttal · Authors · 2023-08-09

Dear Area Chairs and Reviewers,

We appreciate the reviewers’ time, valuable comments, and constructive suggestions. From an overall perspective, we are happy to see that the reviewers approve of the novelty (Co1X, Vi96), originality (Co1X, GN8b), and generality（Co1X, 2ZCa, Vi96) of our approach. In particular, we are grateful for the acknowledgment of the significant improvement compared to the state-of-the-art method (Co1X, Vi96, GN8b), and the reviewers' recognition of our method's contributions to the field (Vi96, GN8b).

Apart from the positive feedback, some concerns from reviewers are in common, so we give the global responses as follows:

> **Q1: Why not adopt an end-to-end model instead of a two-phase model?**

Thanks for your question which is worth for discussion.

First, the SOTA PRNet (Cheng et al, NeurIPS 2022) is an end-to-end approach yet our method notably outperforms it regarding total time cost, wirelength, and overflow quality.

Second, to ensure the connectivity of the whole routing, we design a two-phase routing model which generate hubs and routes. The first phase in our approach includes a discretization process, turning continuous probabilities into determined hubs. These hubs are discrete values and make differentiable learning hardly applicable.

We pursuit the cost-effective end-to-end solver for further work.

> **Q2: The significance of our model.**

First, to our best knowledge, our HubRouter is the first generative model that ensures connectivity in the global routing task. We have conducted several experiments on our approach in comparison with the SOTA PRNet, a generative model that suffers from unconnectivity, which indicates the significance of connectivity in the task.

Second, one of the main contributions of HubRouter is the notable enhancement of inference speed. As shown in Table 3, HubRouter is on average 12x faster than SOTA PRNet. The decline of inference time is meaningful and crucial in the global routing task.

Third, since a theoretical lower bound of this task exists, we further compare the relative error in Table 1 in the rebuttal PDF, where the relative error is computed as $(WL-b)/b$. Here, $b$ denotes the theoretical lower bound. It turns out that the WL promotion of our approach is not marginal compared with the SOTA PRNet.

**A one-page PDF is uploaded that contains corresponding tables and figures in the response.**

In the following, we provide detailed answers. We are glad to give further response for informed evaluation.

---

### Decision · Program_Chairs · 2023-09-21

**Decision:**

Accept (poster)

**Comment:**

Strengths:

The paper presents an innovative approach called HubRouter, designed to tackle the challenge of route unconnectivity in global routing. It introduces a hub generation and hub-pin connection scheme, effectively addressing this issue. Through rigorous experimentation, the results demonstrate HubRouter's superior performance compared to existing generative global routing models, showcasing improvements in wirelength, overflow, and time efficiency. Notably, the approach's versatility extends beyond global routing, as the authors successfully apply it to RSMT construction and interactive path replanning, underscoring its broad applicability.

Weaknesses:

Reviewer 2ZCa raised a pertinent question regarding the paper's clarity in specifying when to use the proposed model in specific situations. Although the authors responded during the rebuttal period by emphasizing the different behaviors of generative algorithms based on industrial needs rather than dataset characteristics, this critical aspect warrants further detailed evaluation and discussion in the paper.

Furthermore, the paper falls short in adequately addressing the question posed about the real-world benefits of achieving a 1% improvement in wirelength (WL). While the authors did assert the significance of this improvement and reiterated correctness rates and inference times, the revised version should directly explain the tangible real-world advantages of the 1% improvement in WL. This clarification would enhance the evaluation of the experimental section and provide a more comprehensive understanding of the practical implications.